# Deep Equilibrium Algorithmic Reasoning

**Dobrik Georgiev**
University of Cambridge
dgg30@cam.ac.uk

**JJ Wilson**
Independent Researcher
josephjwilson74@gmail.com

**Davide Buffelli**
MediaTek Research
davide.buffelli@mtkresearch.com

**Pietro Liò**
University of Cambridge
pl219@cam.ac.uk

## Abstract

Neural Algorithmic Reasoning (NAR) research has demonstrated that graph neural networks (GNNs) could learn to execute classical algorithms. However, most previous approaches have always used a recurrent architecture, where each iteration of the GNN matches an iteration of the algorithm. In this paper we study neurally solving algorithms from a different perspective: since the algorithm's solution is often an equilibrium, it is possible to find the solution directly by solving an equilibrium equation. Our approach requires no information on the ground-truth number of steps of the algorithm, both during train and test time. Furthermore, the proposed method improves the performance of GNNs on executing algorithms and is a step towards speeding up existing NAR models. Our empirical evidence, leveraging algorithms from the CLRS-30 benchmark, validates that one can train a network to solve algorithmic problems by directly finding the equilibrium. We discuss the practical implementation of such models and propose regularisations to improve the performance of these equilibrium reasoners.

## 1 Introduction

Algorithms, while straightforward in theory, become challenging to deploy in real-world scenarios. They operate in abstract domains with very specific conditions and types of inputs, which are represented with scalars. The main hurdle is the need to "collapse" reality into a scalar every time an algorithm is used, something usually done based on intuition rather than principled science [25]. Neural Algorithmic Reasoning (NAR; 46) has been proposed to address this issue by utilising specialised neural network architectures to break this scalar bottleneck by executing algorithms in higher-dimensional space made out of arrays of numbers (vectors). The task of mapping reality into this vectorial space is delegated to automated gradient-based optimisation techniques rather than relying on human operators.

While NAR does not provide the correctness guarantees of its classical counterparts, robust performance can be achieved through *alignment* [55] – submodules of the model architecture correspond to easy-to-learn subparts of the algorithm (or class of). Graph Neural Networks (GNNs) have emerged as the most convenient architecture to execute all types of algorithms [29] and GNNs that align better to the target algorithm achieve better generalisation. This alignment game [50] has led to a sequence of exciting research – from aligning the architecture with iterative algorithms [42] to proving that "graph neural networks are dynamic programmers" [13], especially if their message-passing function [21] takes into account 3-node interactions.

---

[0]Source code available here: `https://github.com/HekpoMaH/DEAR`

38th Conference on Neural Information Processing Systems (NeurIPS 2024).

The aforementioned papers focus on aligning the computation of the GNN with an algorithm or a specific algorithm class (e.g. dynamic programming), but ignore the properties *at the time of algorithm termination*. For the algorithms in the CLRS-30 algorithmic reasoning benchmark [45] once the optimal solution is found, further algorithm iterations will not change it. For example, in dynamic programming shortest-paths algorithms making additional iterations would not alter the optimality of the shortest paths' distances found. Such a state is called an *equilibrium* – additional applications of a function (an algorithm's iteration) to the state leave it unchanged.

In this paper:

1. We explore the connection between execution of algorithms and equilibrium finding through the use of Denotational semantics and Domain theory. (section 3)

2. Inspired by the above, we implement a class of deep equilibrium algorithmic reasoners (DEARs) that learn algorithms by identifying the equilibrium point of the GNN equation and propose improvements to them. (section 4)

3. Our results suggest that the above reasoners can be *successfully* trained. Not only does equilibrium algorithmic reasoning achieve better overall performance with less expressive (and expensive) GNNs, but is also competitive to the more expressive (and expensive) NAR models. All this is done without providing any information on the number of algorithmic steps – neither at train nor at test time. (section 5)

4. DEARs also drastically improve the inference speed – an achievement made possible by the use of optimised root-finding algorithms and by decoupling the neural model from the *sequential* implementation of algorithms in standard benchmark datasets. (section 5)

**Related work**   The main proposed application of NAR is settings where one wants to apply an algorithm, but it is impossible to represent reality with a single scalar, hence an "executor" operating in vector space and faithful to the algorithm is required [47, 50]. As NAR models are neural clones of algorithms, they need to provide correct output even for previously unobserved input sizes. Achieving robust out-of-distribution (OOD) generalisation is tricky. To this end a plethora of works have dedicated their attention to it – [8, 13, 14, 42] to name a few. Except for one concurrent work (a blog post;[53]), those works focus on improving the GNN step and many completely ignore the termination of algorithms or any properties of the last state, such as equilibrium. This work, similarly to Xhonneux et al. [53], studies neurally finding solutions to algorithms by relying on the equilibrium property. We, however, attempt to give the precise assumptions required for this approach to work. Further, we implement a more robust model than theirs, which achieves comparable or better performance to baselines. Finally, we propose modifications to improve equilibrium NARs.

## 2   Background

**Algorithmic Reasoning**   Let $A : \mathbb{I}_A \to \mathbb{O}_A$ be an algorithm, acting on some input $\boldsymbol{x} \in \mathbb{I}_A$, producing an output $A(\boldsymbol{x}) \in \mathbb{O}_A$ and let $\mathbb{I}_A/\mathbb{O}_A$ be the set of possible inputs/outputs $A$ can read-/return. In algorithmic reasoning, we aim to learn a function $\mathcal{A} : \mathbb{I}_A \to \mathbb{O}_A$, such that $\mathcal{A} \approx A$. Importantly, we will not be learning simply an input-output mapping, but we will aim to align to the algorithm $A$'s trajectory. The alignment is often achieved through direct supervision[1] on the intermediate states of the algorithm. To capture the execution of $A$ on an input $x$ we can represent it as

$$\bar{\boldsymbol{h}}_0 = \text{PREPROC}(\boldsymbol{x}) \ \ (1\text{a}) \qquad \bar{\boldsymbol{h}}_\tau = \underbrace{A_t(\dots A_t(\bar{\boldsymbol{h}}_0)\dots)}_{\tau \text{ times}} \ \ (1\text{b}) \qquad A(\boldsymbol{x}) = \text{POSTPROC}(\bar{\boldsymbol{h}}_\tau) \ \ (1\text{c})$$

where PREPROC and POSTPROC are some simple pre- and post-processing (e.g.initialising auxiliary variables or returning the correct variable), $\bar{\boldsymbol{h}}_\tau \in \mathbb{H}_A$ is $A$'s internal (**h**idden) state, $A_t$ is a subroutine (or a set of) that is executed at each step and the number of steps depends on a boolean property being satisfied (e.g. all nodes visited). It is therefore no surprise that the encode-process-decode architecture [24] is the de-facto choice when it comes to NAR. Thus, the architecture can be neatly represented as a composition of three learnable components: $\mathcal{A} = g_\mathcal{A} \circ P \circ f_\mathcal{A}$, where $g_\mathcal{A} : \mathbb{I}_A \to \mathbb{R}^d$

---

[1]Recent research [8, 37] has shown that alternative, causality-inspired, ways of alignment also exist.

and $f_{\mathcal{A}} : \mathbb{R}^d \rightarrow \mathbb{O}_A$ are the encoder and decoder function respectively (usually linear projections) and $P : \mathbb{R}^d \rightarrow \mathbb{R}^d$ is a processor that mimics the rollout (Equation 1b) of $A$. The processor often uses a message-passing GNN at its core.

**CLRS-30** The *CLRS-30* benchmark [45] includes 30 iconic algorithms from the *Introduction to Algorithms* textbook [11]. Each data instance for an algorithm $A$ is a graph annotated with features from different algorithm stages (*input*, *output*, and *hint*), each associated with a location (*node*, *edge*, and *graph*). Hints contain time series data representing the algorithm rollout and include a temporal dimension often used to infer the number of steps $\tau$. Features in CLRS-30 have various datatypes with associated losses for training. The test split, designed for assessing out-of-distribution (OOD) generalisation, comprises graphs four times larger than the training size.

**Deep Equilibrium Models** Deep equilibrium models [DEQs 4] are a class of implicit neural networks [20]. The functions modelled with DEQs are of the form:

$$\boldsymbol{z}^* = f_\theta(\boldsymbol{z}^*, \boldsymbol{x}) \tag{2}$$

where $\boldsymbol{x}$ is input, $f_\theta$ is a function parametrised by $\theta$ (e.g. a neural network) and $\boldsymbol{z}^*$ is the output. $\boldsymbol{z}^*$ is an equilibrium point to the eventual output value of an infinite depth network where each layer's weights are shared, i.e. $f_\theta^{[i]} = f_\theta$. By re-expressing (2) as $g_\theta = f_\theta(\boldsymbol{z}^*, \boldsymbol{x}) - \boldsymbol{z}^*$ DEQs allow us to find the fixed point $\boldsymbol{z}^*$ via any black-box root-finding method [e.g. 2, 9], without the actual need to unroll the equation until convergence, allowing us to reduce steps. For backpropagation the gradient $\partial \mathcal{L}/\partial \theta$ could be calculated using the Implicit Function Theorem (cf. 4) and no intermediate state has to be stored, giving us a constant memory cost of gradient computation regardless of the number of iterations until convergence.

**Expander graphs** MPNNs operate by exchanging information between adjacent nodes [21]. It has been identified that the message passing process can be hindered by a phenomenon known as *oversquashing* [1], which occurs when a large volume of messages are compressed into fixed-sized vectors. The importance of overcoming the negative implication posed by this phenomenon is crucial for the overall expressivity of GNNs [22], particularly in the context of long-range node interactions.

To this end, several spectral methods have been proposed to mitigate oversquashing by increasing the Cheeger constant [3, 6, 30]. A higher Cheeger constant provides a measurement that a graph is globally lacking bottlenecks. The novel approaches include graph rewiring techniques [7, 44], as well as significant independent bodies of research recognising the efficacy of expander graphs [6, 12, 41], due to their desirable properties.

Expander graphs are proven to be highly connected sparse graphs ($|E| = \mathcal{O}(|V|)$) with a low diameter [35], thus offering advantageous properties for information propagation. Consequently, this facilitates messages to be passed between any pair of nodes in a short number of hops, and as a result, alleviating oversquashing. Formally, a graph $G = (V, E)$ is defined as an expander if it satisfies certain expansion properties. One common definition involves the aforementioned Cheeger constant. In the work of Deac et al. [12], a high Cheeger constant is equivalent to a graph being bottleneck free [12, Definition 3], and that an expander has a high Cheeger constant [12, Theorem 5].

There are various methods for constructing expander graphs. We opt for the *deterministic* algebraic approach as in Deac et al. [12], utilising Cayley graphs. Specifically, we leverage Definition 8 and Theorem 9 of [12] to construct the Cayley graph for the *special linear group* $\mathrm{SL}(2, \mathbb{Z}_n)$ and its generating set $S_n$ – see p.5 of Deac et al. [12] for details. Note, that the order of a Cayley graph for $\mathbb{Z}_n$ is $|V| = \mathcal{O}(n^3)$. Hence, for many input graphs, a Cayley graph of the same size may not exist.

## 3 Denotational semantics: The denotation of a while loop statement

This aim of this section is to draw the parallel between finding equilibriums and executing an algorithm, in order to answer if and when an equilibrium NAR model can be successful. The following paragraphs introduce the mathematical tools for formalising *fixed points* – denotational semantics [39] and domain theory [38].

**Denotational semantics** To every programming language expression[2] $P$ denotational semantics provides an interpretation $[\![P]\!]$, which is a mathematical object (often a function), representing the behaviour of the expression to different inputs. These *denotations* must be: 1) abstract, i.e. independent of language and hardware, hence functions are natural choice; 2) compositional, i.e. $[\![P]\!]$ can only be defined in terms of the *denotations* of $P$'s subexpressions, but not $P$ itself; 3) model the computation $P$ performs. As an example, we will use the lightweight imperative programming language **IMP**[3]. It consists of *numbers*, *locations*, *arithmetic expressions*, *boolean expressions* and *commands*. Examples of **IMP** are given in Appendix A – we will use blue for **IMP** and encourage the reader to check how we use colour in the appendix. Albeit small, the language is Turing-complete and all algorithms we experiment with can be defined in it.

Denote the set of variable locations with $\mathbb{L}$ – those are all variables/array elements we can ever define. A good analogy to think of is the addresses in the language C. The notation we can use to represent a program state is $[X \mapsto 1, B \mapsto -48, \dots]$ and means that the value of $X$ is 1, the value of $B$ is $-48$ and so on. In other words, program states map locations to integers, s.t. a location can be mapped only once. Hence states are functions and the set of all program states $State$ is the set of functions mapping locations to integers: given $s \in State$, $s(L) \in \mathbb{Z}$ is the value at the location $L$ *for the state* $s$. The value for location $L$ in a different $s' \in State$, $s'(L)$, may or may not differ. The denotation of arithmetic / boolean expressions / commands are the functions with domain $State$. These will be represented in the form of *lambda abstractions*, i.e. $\lambda x \in S.M$ rather than $f(x \in S) = M$, where $S$ is a set and $M$ is a function body. The codomain of the denotation depends on the type of expression: $[\![a]\!] : State \to \mathbb{Z}$ for arithmetic expressions, $[\![b]\!] : State \to \mathbb{B}$, for boolean expressions and $[\![c]\!] : State \rightharpoonup State$ for commands. Since commands transform state, they are also called state transformers. ***Commands' denotations are partial functions, as expressions like*** `while true do skip` ***never terminate and have no denotation.***

For a large portion of the above language, it is trivial and intuitive to define the denotations by structural recursion. For example:

$$[\![\texttt{if } b \texttt{ then } c_0 \texttt{ else } c_1]\!] = \lambda s \in State. \begin{cases} [\![c_0]\!](s) & \text{if } [\![b]\!](s) \text{ is true} \\ [\![c_1]\!](s) & \text{otherwise} \end{cases}$$

$$[\![(c_0 ; c_1)]\!] = \lambda s \in State. [\![c_1]\!]([\![c_0]\!](s)) \qquad [\![skip]\!] = \lambda s \in State.\ s$$

The only denotation that cannot be expressed recursively, is that of the `while` construct. Let $w = \texttt{while } b \texttt{ do } c$. By program equivalence, $w = \texttt{if } b \texttt{ then } (c;w) \texttt{ else skip}$. Therefore

$$[\![w]\!] = [\![\texttt{if } b \texttt{ then } (c;w) \texttt{ else skip}]\!] = \lambda s \in State. \begin{cases} [\![w]\!]([\![c]\!](s)) & \text{if } [\![b]\!](s) \text{ is true} \\ s & \text{otherwise} \end{cases}$$

but this is not a valid definition, since it reuses $[\![w]\!]$ (highlighted in red above). Denotational semantics solves this problem, by defining a function $f_{b,c} : (State \rightharpoonup State) \to (State \rightharpoonup State)$ which takes one state transformer and returns another:

$$f_{b,c} = \lambda \hat{w} \in (State \rightharpoonup State).\lambda s \in State. \begin{cases} \hat{w}(c(s)) & \text{if } b(s) \\ s & \text{otherwise} \end{cases} \tag{3}$$

$\hat{w}$ is now a function variable. The denotation of $[\![w]\!]$ is the fixed point of $f_{[\![b]\!],[\![c]\!]}$, i.e. $[\![w]\!] = f_{[\![b]\!],[\![c]\!]}([\![w]\!])$. In order to find the denotation, we need to solve the fixed point. To aid the reader a full worked example of computing the denotation for a `while` loop construct is given in Appendix B.

**Domain theory** Scott [38] provides a framework with which we can both find and also characterise solutions for fixed point equations.[4] Define $D$ as the domain of state transformers $(State \rightharpoonup State)$. A partial order[5] $\sqsubseteq$ on $D$ is defined as follows: $w \sqsubseteq w'$ iff $\forall s \in State$ if $w(s)$ is defined then $w(s) = w'(s)$. In other words $w'$ keeps old mappings and only defines new ones. The totally undefined partial function $\bot$ is the least element in $D$. This function contains no location to value

---

[2]Note the abuse of notation. For this subsection, we will forget we use $P$ for the neural processor.

[3]Due to space constraints, we omit the formal language definition (Winskel [52], p. 11-13), and we use a condensed version [18] of the denotational syntax, given in Chapter 5 of Winskel [52].

[4]For detailed definitions and proofs, please refer to §5.4 of Winskel [52].

[5]It is reflexive, transitive and anti-symmetric.

mappings. A chain is a sequence of elements of $D$, s.t. $d_0 \sqsubseteq d_1 \sqsubseteq d_2 \sqsubseteq \dots$ . The supremum of the chain, called *least upper bound (lub)*, is denoted as $\bigsqcup_{n \geq 0} d_n$. There could exist different chains, but, by definition, all chains in a domain must have a lub.

A function $f : D \to D$ is monotonic iff $\forall d, d' \in D. \ (d \sqsubseteq d' \Rightarrow f(d) \sqsubseteq f(d'))$. In other words, if the second state defined more mappings than the first and we apply one iteration step to both states, the state resulting from the second will still define more mappings. Monotonic functions for which $\bigsqcup_{n \geq 0} f(d_n) = f(\bigsqcup_{n \geq 0} d_n)$ are also called continuous. In plain language, if a function is continuous and we are provided with a chain, the lub of $f$ applied to every chain element is the same as $f$ applied to the lub of the chain. An element $d'' \in D$ is defined to be *pre-fixed point* if $f(d'') \sqsubseteq d''$ – applying $f$ does not define anything new. The fixed point $fix(f)$ of $f$ is the least pre-fixed point of $f$. By utilising antisymmetry[6] of $\sqsubseteq$ and the two properties of $fix(f)$ (pre-fixed point and least) we can obtain $f(fix(f)) = fix(f)$. By Tarski's theorem [43], any continuous $f : D \to D$ has a least pre-fixed point. This fixed point can be found, by taking the lub of the chain of applications of f: $fix(f) = \bigsqcup_{n \geq 0} f^n(\bot)$. The helper function $f_{b,c}$ from Equation 3 is continuous [proof is given on p.120 of 23], therefore a direct result is that if the `while` $b$ `do` $c$ terminates then its denotation exists and is *the least* fixed point of sequence of iterations (compared to picking any fixed point).

**Denotational semantics and NAR**   The above detour into denotational semantics has affirmed the existence of a connection between equilibrium models and algorithms (as conjectured by Xhonneux et al. [53]). Under the assumptions that:

- the algorithms always terminate – while not computable in the general case, this holds true for experiments, as we are dealing with offline algorithms with provable termination

- the algorithms we train on can be modelled as "`while` $b$ `do` $c$" constructs within **IMP**

the least fixed point exists and can be found by taking the first "state" of the algorithm where future iterations on it have no effect. In Appendix C we have further annotated three algorithms from the official code of the CLRS benchmark: BFS, Floyd-Warshall, strongly connected components. Those annotations clearly show that algorithms can be rewritten in **IMP** regardless of their implementation size. While BFS is clearly a "`while` $b$ `do` $c$"-type of algorithm, annotating the other two reveals that either the network may need more input features to decide termination (Floyd-Warshall; Listing 2) or that the algorithm can be composed of several while loops where each $c$ is another while loop on its own (strongly connected components; Listing 3). Fortunately, our approach is not doomed to fail: a single DEQ layer can model any number of "stacked" DEQ layers [32, chapter 4].

## 4   Deep equilibrium algorithmic reasoning

**Architecture**   We implement our processors/encoders/decoders following Ibarz et al. [29]. The most notable difference[7] to their implementation is that ours uses a sparse graph representation. This requires us to assume a fully connected graph on tasks where no graph structure exists, in order to be able to give pointer predictions, and to reimplement the strongly connected components algorithm so that the output pointers are always in the edge set (this did not change the difficulty of the task).

The final node embeddings, from which the output is decoded, are the solution to the equation:

$$\mathbf{H}^{(*)} = P_{\mathbf{UE}}(\mathbf{H}^{(*)}) \tag{4}$$

where $P_{\mathbf{UE}}(\mathbf{H}^{(*)}) = P(\mathbf{H}^{(*)}, \mathbf{U}, \mathbf{E})$, $\mathbf{U}/\mathbf{E}$ are the encoded node and edge feature matrices and $P$ is the processor function. $\mathbf{H}^{(t)}$ are the stacked latent states of the nodes at timestep $t$ (with $\mathbf{H}^{(0)} = \mathbf{0}$). The above Equation 4 matches the signature of Equation 2, and can be solved via root-finding (we employ `torchdeq` [19]; MIT License), as if it is $f_\theta$ of a DEQ. Any model using this technique will be called *deep equilibrium algorithmic reasoner* (**DEAR**) in our experiments. The default processor in the majority of our experiments is a PGN [48] with a gating mechanism as in Ibarz et al. [29], but we note that DEARs can use any kind of processor.

---

[6]$a \sqsubseteq b$ and $b \sqsubseteq a$ implies $a = b$

[7]See Appendix D for others not mentioned in the main text

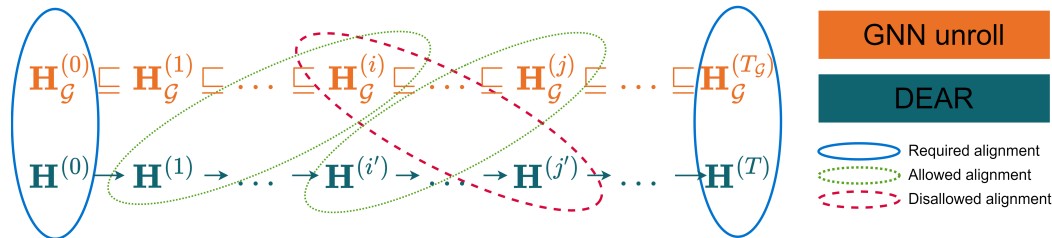

Figure 1: Proposed alignment rule: every state in the DEAR trajectory should "go forward". Alignments to a GNN state that has already been "passed" are disallowed. First and last states must always align. We intentionally use arrows instead of $\sqsubseteq$ for DEAR, as $\sqsubseteq$ may not hold for DEAR's trajectory.

**Finding the fixed point**   The `torchdeq` library provides several solvers. The most basic one is *fixed-point iteration*, equivalent to repeating Equation 4 until convergence. However, in our very first experiments the solver needed more iterations than the algorithm we train on. We therefore opted for the *Anderson* solver (implements Anderson acceleration [2]) and abandoned fixed-point iteration:

$$\hat{\mathbf{H}}^{(t+1)} = P_{\mathbf{UE}}(\mathbf{H}^{(t)}) \qquad \mathbf{H}^{(t+1)} = SolverStep\left(\left[\mathbf{H}^{(0)} \dots \mathbf{H}^{(t)}\right], \hat{\mathbf{H}}^{(t+1)}\right)$$

The criteria for pre-fixed point check `torchdeq` implements is distance-based: for a state $\mathbf{H}^{(t)}$ to be considered a pre-fixed point, the distance to the next state has to be under a pre-defined threshold $\delta$. We kept the criteria but modified the library to always return the least pre-fixed point (see Appendix E). This is in contrast to picking the pre-fixed point with the least distance to next state (the default option in `torchdeq`) and is a decision largely motivated from section 3. Due to a lack of a suitable definition for the domain of NAR trajectories, we define $\forall t. \mathbf{H}^{(t)} \sqsubseteq \mathbf{H}^{(t+1)}$, i.e. we pick the first state that passes the pre-fixed point check.

**Globally propagating information**   For problems defined on graphs it is in theory possible that the number of solver iterations needed to find equilibrium is less than the diameter of the graph. While, in practice, this is unlikely to happen we hypothesise that improving long-range interactions could improve the convergence of DEAR. For this reason, we adopt the implementation of Cayley Graph Propagation (CGP) [51]. Contrasted to Expander Graph Propagation (EGP) [12], which addresses the graph size misalignment (see section 2) by truncating the Cayley graph, CGP keeps the extra nodes as virtual nodes. The CGP model upholds the aforementioned advantageous properties of an expander graph in a more grounded manner by preserving the complete structure.

In GNNs, the benefits of augmenting a graph with virtual nodes and providing message-passing shortcuts have been observed to improve performance in various tasks [10, 26, 27]; further supported by the theoretical analysis [28]. Additionally, by retaining the complete Cayley graph structure we improve the structure-aware representations by varying the neighbourhood ranges [54].

**No hint by default**   We do not make any use of hints (supervising on intermediate algorithm state). First, although it may seem counterintuitive, it has been shown that a NAR model can successfully generalise, and even give better results when trained to only predict the correct output [8, 37]. Second, the fact that the solver uses the GNN exactly once per call *does not imply that one step of the solver would correspond to one iteration of the algorithm*, bringing uncertainty which DEAR states to match to which algorithm step. While we propose an alignment scheme (see next paragraph), which has the potential to integrate hints, we leave this for future work.

**Alignment**   Our idea of alignment is visualised in Figure 1. We are given two trajectories of states, one obtained from unrolling GNN iterations as in classical NAR and another obtained from using DEAR. We would like to match DEAR to NAR, such that $\forall i \leq j, i' \leq j'$ if we have committed to aligning DEAR state $\mathbf{H}^{(i')}$ to NAR state $\mathbf{H}_{\mathcal{G}}^{(j)}$, we cannot align any $\mathbf{H}^{(j')}$ to $\mathbf{H}_{\mathcal{G}}^{(i)}$ and from the same start we would like to reach the same final state. In other words, *skipping states is allowed, going back in time is not*. This auxiliary supervision would also improve the monotonicity of DEARs, encouraging faster convergence. Enforcing this alignment is done by using an auxiliary loss. Choosing the $L_2$ norm as a distance metric, we use a dynamic programming algorithm (Appendix F)

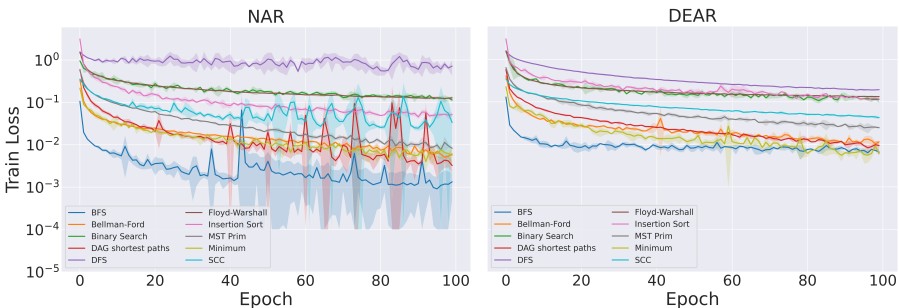

Figure 2: Despite converging to slightly higher train loss our models remain stable during optimisation

to compute the most optimal alignment (normalised by the number of solver steps, in order not to penalise longer trajectories) and supervise on that value.

Even with normalisation, the alignment sometimes had the effect of making the optimisation stuck in local minima where the number of steps to hit equilibrium was as low as 2 and the gradients were uninformative. We combated this in two ways: 1) instead of using the default layer normalisation we switched to GRANOLA [15]; 2) since $f(fix(f)) = f$, we performed a random number of additional iterations [33] and take the last state. The probability of doing $s$ extra steps is $0.5^s$.

## 5    Evaluation

**Setup**    For each algorithm we generated $10^5/100/100$-sized training/validation/test datasets. Training sample sizes vary between 8 and 16 elements (uniformly randomly chosen) validation samples are of size 16. As is standard in NAR literature, we measure the test performance out-of-distribution, so our test samples are of size 64. For algorithms on graphs we generate Erdős–Rényi graphs [17] with edge probabilities $p$ uniformly sampled from the interval $[0.1, 0.9]$, with increments of $0.1$, which is the data distribution our baselines [8, 29] have used. We obtained the ground truth execution trajectories and targets using the CLRS-30 implementation [45].

In our experiments the models have a latent dimensionality of 128, the batch size is 32, the learning rate is $3 \times 10^{-4}$ and we use the Adam optimizer [31]. We train our algorithmic reasoners for 100 epochs, choosing the model with the lowest *task* validation loss (discounting any regularisation; focusing on performance only) for testing. Each task is independently learned, minimising the output loss (losses depend on the algorithm, cf. CLRS-30) plus any regularisation losses. Unless otherwise specified, DEARs employ the Anderson root-finding method from the `torchdeq` library and include Jacobian regularization [5], the tolerance for fixed point criteria on the forward pass is $\delta = 0.1$ (and $\frac{\delta}{10}$ on the backwards) and is based on the relative $L^2$ norm between GNN states. Standard deviations are based on 3 seeds. If run on a single 4090 GPU one would need about 3 weeks of *total* compute.

The performance metric we measure is the out-of-distribution accuracy, hence the larger test instances. The definition of accuracy varies between algorithms and is based on the specification of the algorithm itself. We refer the reader to Veličković et al. [45] and CLRS-30 for corresponding accuracy metrics definitions. The main baselines we compare against are the results *reported* by Xhonneux et al. [53], as no implementation is publicly available, and a NAR architecture with a PGN processor trained in the no-hint regime, as done by Bevilacqua et al. [8]. As, logically, models that are provided the ground-truth number of steps at test time will perform better, we also add as additional baselines a model that always uses 64 steps at test time and a model that has a dedicated network to decide termination [49]. In order to understand how we compare to other, architectural alignments, we also provide a comparison with a more expressive processor (Triplet-MPNN).

### 5.1    Results

We present results for 10 key algorithms (most of the ones used in Bevilacqua et al. [8]) in Table 1.

**DEARs are reasoners**    The first set of experiments aims to establish whether learning to execute algorithms by finding the least fixed point is possible. As Xhonneux et al. [53] report that their models

Table 1: Test accuracy for different algorithms and models. Models with a diamond ($\diamond$ or $\blacklozenge$) iterate for the correct amount of steps during train time (may differ between datapoints). Filled diamond ($\blacklozenge$) means the ground truth number of steps is also given at test time. LT stands for **l**earnt **t**ermination – the model that uses a termination network. For DEM [53] we leave a $-$ when no results are reported and we report two results for shortest path and MST as it is unclear to us from the main text how they differentiated between the two. We do not run DEAR with CGP for array tasks as they operate on fully-connected graphs.

| Algorithm | NAR$^\blacklozenge$ | NAR$^\blacklozenge$ (Triplet-MPNN) | NAR$^\diamond$ (LT) | DEM | DEAR (ours) | DEAR (with CGP; ours) |
|---|---|---|---|---|---|---|
| **Weighted graphs** | | | | | | |
| **Bellman-F.** | $97.06\% \pm 0.40$ | $97.23\% \pm 0.15$ | $95.39\% \pm 1.42$ | $96.4\%/78.8\%$ | $96.78\% \pm 0.43$ | $94.23\% \pm 0.59$ |
| **Floyd-W.** | $52.53\% \pm 0.98$ | $61.86\% \pm 1.57$ | $49.30\% \pm 0.53$ | - | $55.75\% \pm 2.20$ | $53.20\% \pm 2.45$ |
| **DSP** | $94.21\% \pm 1.77$ | $93.32\% \pm 1.60$ | $88.30\% \pm 1.04$ | - | $89.81\% \pm 0.14$ | $89.49\% \pm 0.17$ |
| **MST Prim** | $93.56\% \pm 0.77$ | $92.01\% \pm 1.50$ | $87.69\% \pm 1.17$ | $82.3\%/75.2\%$ | $88.67\% \pm 0.74$ | $86.37\% \pm 0.36$ |
| **Unweighted graphs** | | | | | | |
| **BFS** | $99.85\% \pm 0.09$ | $99.69\% \pm 0.29$ | $99.51\% \pm 0.06$ | $53.8\%$ | $98.73\% \pm 0.37$ | $98.28\% \pm 0.55$ |
| **DFS** | $16.89\% \pm 5.73$ | $31.20\% \pm 4.02$ | $29.07\% \pm 2.32$ | $5.0\%$ | $40.62\% \pm 0.44$ | $23.87\% \pm 2.49$ |
| **SCC** | $40.70\% \pm 1.39$ | $46.84\% \pm 1.70$ | $39.33\% \pm 1.52$ | - | $43.63\% \pm 1.19$ | $38.71\% \pm 0.45$ |
| **Arrays** (assumes fully-connected graph) | | | | | | |
| **Search** (Binary) | $94.67\% \pm 2.31$ | $93.33\% \pm 2.31$ | $84.33\% \pm 8.33$ | - | $59.00\% \pm 12.3$ | - |
| **Minimum** | $97.67\% \pm 5.73$ | $96.67\% \pm 2.31$ | $94.00\% \pm 2.00$ | - | $97.22\% \pm 3.82$ | - |
| **Sort** (Ins.) | $27.07\% \pm 10.3$ | $63.67\% \pm 39.97$ | $33.8\% \pm 12.04$ | - | $86.93\% \pm 3.87$ | - |
| **Overall** | $71.42\%$ | $\mathbf{77.58}\%$ | $70.07\%$ | - | $\underline{75.42\%}$ | - |

were prone to optimisation issues, we first compared the training loss for a standard NAR model and a DEAR model with the same neural components. The plots are visualised in Figure 2. In line with the previous work, we observed that the DEAR tends to converge to a slightly higher training loss as no algorithm's mean training loss dropped below 0.01. However, as evident in Figure 2, we found the optimisation to be overall stable, and the final train loss difference between NAR and DEAR was never greater than 0.1 – see Appendix G. We are unaware if Xhonneux et al. [53] observed the same numerical differences, but we were overall satisfied with the convergence of DEARs.

**Equilibrium is a useful inductive bias**   DEAR outperforms both of the above baselines achieving a 4-5% overall score increase, suggesting that aligning to the equilibrium property is a useful inductive bias. Moreover, DEAR with a PGN processor is comparable to a NAR with the more expressive Triplet-MPNN, achieving only 2% lower overall accuracy. This commendable achievement required no information about the ground-truth number of steps neither at train time nor during inference. A more detailed performance analysis follows.

On weighted graph algorithms our model performed on par with the baseline NAR no-hint model for Bellman-Ford, it outperformed the baseline on Floyd-Warshall, and scored slightly behind on the other two. On unweighted ones, it retained fairly high BFS accuracy compared to DEM and it provided better scores for DFS and Strongly Connected Components (SCC). Unfortunately, even though for this kind of algorithms we used separate edge features for the CGP, in order to distinguish CGP edges from input ones, CGP had a detrimental effect. We hypothesise that algorithms like DFS and SCC need a more advanced architecture or require different task specifications (the algorithm for SCC has a parallel "twin"; see [16]) in order to generalise OOD. On algorithms on arrays, we got a significant performance improvement in the sorting task and got almost equal scores for min finding. However, the model underperformed by a large margin on the binary search task (in red). This result was very concerning, so we investigated further – Appendix H showed that DEARs overfitted a lot on the classic representation of binary search and that when the task is designed carefully, DEARs can reach overall performance of a Triplet-MPNN NAR architecture.

**Effects of using CGP**   Despite the slightly lower accuracies, our experiments with CGP have not been futile. In Figure 3, we observe that for almost half of the algorithms CGP applies to, it had a positive effect on the loss convergence – 3/7 algorithms converged to at least half an order of magnitude lower train loss. The rest did not experience any strong negative effects of CGP. Per-algorithm plots can be found in Appendix I. We believe that the reduced accuracies are due to the

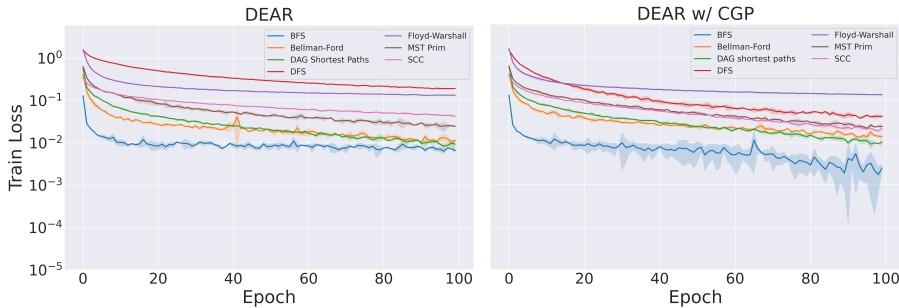

Figure 3: Cayely graph propagation can help with convergence

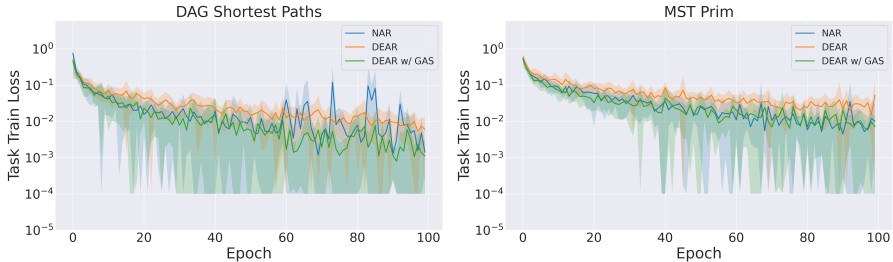

Figure 4: Alignment (with GRANOLA and stochasticity; **DEAR w/ GAS**) gives better convergence

nearest Cayley graph for the training sizes being unique and a size of 24 nodes. Our deterministic approach of generating a fixed Cayley graph for CGP, whose size is still distinct from test-time size leads to overfitting; what we may observe here. Future avenues of work may want to investigate this by methodically removing the Cayley graph's edges, but still retaining the desirable expansion properties [6], or by exploring alternative novel graph wiring techniques [7]. However, the limitation of these proposed approaches in comparison to CGP is that they may require *dedicated preprocessing* to scale (one of the desirable criteria set by the EGP method), therefore providing an interesting line of future work.

**Alignment can distill knowledge into DEARs** For evaluating our alignment we focused on the non-CGP version of DEAR and decided to pick algorithms where: 1) The baseline performs reasonably well (90+% accuracy), so as to provide good support; 2) the DEAR underperforms substantially. The algorithms to fulfil those requirements are: DAG Shortest paths, MST Prim and Binary Search.

Results are presented in Table 2. At first glance, the only algorithm that substantially improved was binary search, giving an almost 20% increase. The final test accuracy, however, does not represent all reality: Figure 4 shows that the task train loss (loss excluding any regularisers) for the model with alignment decreases, compared to no alignment and reaches similar levels as the one observed

Table 2: Test accuracy with and without alignment.

|  | **DSP** | **MST-Prim** | **Binary Search** |
|---|---|---|---|
| **NAR** | $94.21\% \pm 1.77$ | $93.56\% \pm 0.77$ | $94.67\% \pm 2.31$ |
| **DEAR** | $89.81\% \pm 0.14$ | $88.67\% \pm 0.74$ | $59.00\% \pm 12.3$ |
| **DEAR** (alignment) | $89.65\% \pm 2.95$ | $90.37\% \pm 1.19$ | $77.33\% \pm 4.51$ |

for the non-DEQ solution. So, is it overfitting again? We argue it is not. Figure 5 shows that the *test (OOD)* accuracy per epoch increases when using alignment, reaching similar accuracies to NAR for the DAG shortest path problem and improving over plain DEAR for MST-Prim, suggesting that choosing the right model using validation seed is hard in NAR [34]. Lastly, we would like to note that: 1) although GRANOLA+stochasticity does bring benefits on its own, alignment is necessary to narrow the gap to the NAR training loss (Appendix J); 2) We never reached perfect (0) alignment loss, suggesting better alignment techniques may further boost performance.

**DEARs are highly parallel** DEAR is not bound to follow sequential trajectories and GNNs are more aligned to parallel algorithms than to sequential ones [16]. As the cost for one step of DEAR (GNN iteration + solver) is at least as high as one step of an NAR model (GNN iteration only), we

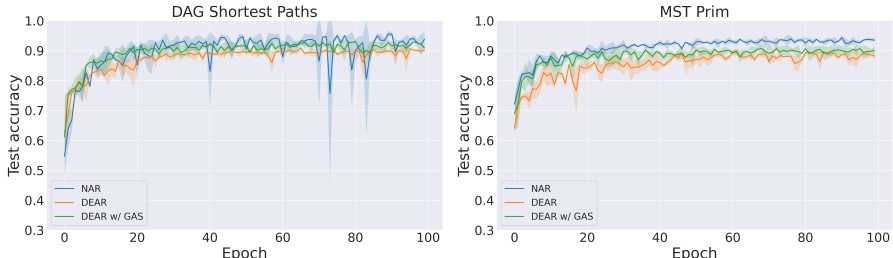

Figure 5: Alignment (**DEAR w/ GAS**) leads to improvements OOD

Table 3: Mean inference time in seconds per sample. Measured on an RTX 4090 GPU. Up/down arrows denote improvements/deteriorations. $\approx$ is used when difference is negligible. A double symbol is used for substantial ($5\times$) differences.

| | Bellman-F. ↑ | Floyd-W. ↑ | DSP ↓ | MST Prim ↓ | BFS ↑ | DFS ↓↓ | SCC ↓↓ | Search (Binary) ≈ | Minimum ↓ | Sort (Insertion) ↓↓ |
|---|---|---|---|---|---|---|---|---|---|---|
| **NAR**♦ | 0.0118 | 0.0916 | 0.1334 | 0.0708 | 0.0094 | 0.2440 | 0.4017 | 0.0125 | 0.0684 | 0.5680 |
| **DEAR** | 0.0215 | 0.1102 | 0.0345 | 0.0297 | 0.0137 | 0.0478 | 0.0253 | 0.0131 | 0.0174 | 0.0260 |

Table 4: DEAR is architecture invariant and can also run with a Triplet-MPNN processor.

| | Floyd-W. | DFS | SCC | Search (Parallel) | Sort | Overall |
|---|---|---|---|---|---|---|
| **NAR**♦ | $61.86\% \pm 1.57$ | $31.20\% \pm 4.02$ | $\mathbf{46.84\% \pm 1.70}$ | $\mathbf{93.33\% \pm 0.58}$ | $63.67\% \pm 39.97$ | $59.18\%$ |
| **DEAR** (ours) | $\mathbf{62.29\% \pm 2.71}$ | $\mathbf{42.73\% \pm 2.79}$ | $45.12\% \pm 1.52$ | $87.00\% \pm 5.57$ | $\mathbf{82.34\% \pm 9.46}$ | $\mathbf{63.90\%}$ |

used the inference speed of a DEAR as a measure of how parallel the final learnt algorithm is. Results are presented in Table 3. An immediate observation is that DEAR improves inference times across almost all algorithms. The only ones that were executed slower are: 1) Bellman-Ford and BFS, which are highly parallelised in the CLRS-30 implementation, so an improvement on them was unlikely; 2) Floyd-Warshall where the difference, although present, is marginal and we account it to the added overhead from the solver; 3) Binary search, where performance was almost identical. These results suggest that although not always guaranteed (the case for searching), it is very likely that a DEAR will learn a parallel algorithm. The most substantial improvements, in line with our past observations in Engelmayer et al. [16], were on the tasks of sorting and strongly-connected components.

**DEARs are foundational**   Up until this point, DEAR was run with a PGN processor, which is a lightweight, yet well-performant NAR processor architecture. The last set of experiments aims to show that equilibrium reasoning is not tied to only one type of processor/architecture. It is rather a **class of models/foundational model** as it can natively support different types of processors. To verify this claim, we present results with DEAR using the Triplet-MPNN architecture in Table 4. As Triplet-MPNN is computationally expensive, we tested algorithms for which NAR with Triplet-MPNN improves over NAR with PGN. Results indeed confirm that we are not limited to a single type of processor with DEAR, and, as expected, the best overall performance is achieved when using DEAR with the more expressive, Triplet-MPNN, processor.

## 6   Conclusion

Our investigations with equilibrium models have shown that it is possible and even beneficial to merge NAR and DEQs. While our models attained very competitive performance, there are certain limitations that need to be addressed: 1) Better algorithms for alignment can help close the gap even further for Prim's algorithm and binary search; 2) Better model selection is needed in order to know which DEARs would perform well OOD; 3) Graph rewiring techniques may be needed to prevent overfitting with CGP; 4) Algorithmic-aligned criteria for fixed-point may boost OOD generalisation for sequential algorithms. The last point is motivated by the fact that for each step, these algorithms update only a few nodes in the graph, keeping the rest untouched.

## Acknowledgements

Dobrik Georgiev would like to acknowledge the financial support from G-Research towards covering his travel costs.

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

# A IMP: Definitions and examples

Anything highlighted in `blue` below, is part of the **IMP** language. Subexpressions, to avoid confusion, are not coloured.

**IMP** consists of:

- *numbers* – 1, -20, 13930

- *locations* – `AVariableName`, `AnotherVar`, `ArrayName`[11]

- *arithmetic expressions* – `(5+4)*3`, but also `A*55`. Note how variables can be parts of arithmetic expressions.

- *boolean expressions* – `true`, `false`, but also `X==0`, `3*A<B` and `X==0` $\wedge$ `3*A<B`. Note how boolean expressions can be made by using variables or using boolean logic on sub-expressions.

- *commands* – Commands can be one of:

  - `skip`, which is a no-op

  - `X:=a`, where X is assigned the value of arithmetic expression $a$. An example $a$ is `3*B+C`

  - `if` $b$ `then` $c_0$ `else` $c_1$ where $b$ is a boolean expression and $c_0$, $c_1$ are commands. For example `if A < B then C:=0 else C:=A-B` which sets `C` to the difference of A and B, if it is positive

  - $(c_0 ; c_1)$

  - `while` $b$ `do` $c$ which is a while loop repeating command $c$ as long as boolean expression $b$. A (classic) example is `(Y:=1; while X > 0 do (Y:=X*Y; X:=X-1))`, which finds the factorial of X and saves it in Y.

# B Fixed point of a while loop – example

Below, we will denote the state as $[A \mapsto a, B \mapsto b, \dots]$. It means that the value of variable $A$ is $a$ and so on.

Consider the facorial example from above removing the explicit set of Y to 1. To find the denotation $[\![w]\!] = [\![$`while X > 0 do (Y:=X*Y; X:=X-1)`$]\!]$, we first define our $f_{b,c}$

$$f_{b,c}(w)([X \mapsto x, Y \mapsto y]) = \begin{cases} w\left([X \mapsto x-1, Y \mapsto y*x]\right) & \text{if } X > 0 \\ [X \mapsto x, Y \mapsto y] & \text{otherwise} \end{cases}$$

The equivalent definition if we were to keep the lambdas from the original text is

$$f_{b,c} = \lambda w \in (State \rightharpoonup State).\lambda s \in State. \begin{cases} w\left([X \mapsto x-1, Y \mapsto y*x]\right) & \text{if } X > 0 \\ [X \mapsto x, Y \mapsto y] & \text{otherwise} \end{cases}$$

but we will work with the first definition as it is more compact.

The approximations of $f_{b,c}^n$ starting from $f_{b,c}^0 = \bot$ are:

$$f_{b,c}^1 = f_{b,c}(f_{b,c}^0)([X \mapsto x, Y \mapsto y]) \quad =$$

$$= f_{b,c}(\bot)([X \mapsto x, Y \mapsto y]) \quad = \begin{cases} \bot\,([X \mapsto x-1, Y \mapsto y*x]) & \text{if } x > 0 \\ [X \mapsto x, Y \mapsto y] & \text{otherwise} \end{cases}$$

$$= \begin{cases} \text{undefined} & \text{if } x > 0 \\ [X \mapsto x, Y \mapsto y] & \text{otherwise} \end{cases}$$

$$f_{b,c}^2 = f_{b,c}(f_{b,c}^1)([X \mapsto x, Y \mapsto y]) \quad = \begin{cases} f_{b,c}^1\,([X \mapsto x-1, Y \mapsto y*x]) & \text{if } x > 0 \\ [X \mapsto x, Y \mapsto y] & \text{otherwise} \end{cases}$$

$$= \begin{cases} \text{undefined} & \text{if } x - 1 > 0 \\ [X \mapsto x-1, Y \mapsto y*x] & \text{if } x - 1 = 0 \\ [X \mapsto x, Y \mapsto y] & \text{if } x \le 0 \end{cases}$$

$$= \begin{cases} \text{undefined} & \text{if } x > 1 \\ [X \mapsto 0, Y \mapsto y] & \text{if } x = 1 \\ [X \mapsto x, Y \mapsto y] & \text{if } x \le 0 \end{cases}$$

$$f_{b,c}^3 = f_{b,c}(f_{b,c}^2)([X \mapsto x, Y \mapsto y]) \quad = \begin{cases} f_{b,c}^2\,([X \mapsto x-1, Y \mapsto y*x]) & \text{if } x > 0 \\ [X \mapsto x, Y \mapsto y] & \text{if } x \le 0 \end{cases}$$

$$= \begin{cases} \text{undefined} & \text{if } x - 1 > 1 \\ [X \mapsto 0, Y \mapsto y*x] & \text{if } x - 1 = 1 \\ [X \mapsto x-1, Y \mapsto y] & \text{if } x - 1 \le 0 \\ [X \mapsto x, Y \mapsto y] & \text{if } x \le 0 \end{cases}$$

$$= \begin{cases} \text{undefined} & \text{if } x > 2 \\ [X \mapsto 0, Y \mapsto y*2] & \text{if } x = 2 \\ [X \mapsto 0, Y \mapsto y] & \text{if } x = 1 \\ [X \mapsto x, Y \mapsto y] & \text{if } x \le 0 \end{cases}$$

$$\vdots$$

which for $n$ is:

$$f_{b,c}^n = \begin{cases} \text{undefined} & \text{if } x \ge n \\ [X \mapsto 0, Y \mapsto y*(x!)] & \text{if } 0 < x < n \\ [X \mapsto x, Y \mapsto y] & \text{if } x \le 0 \end{cases}$$

The sequence obeys $f_{b,c}^0 \sqsubseteq f_{b,c}^1 \sqsubseteq f_{b,c}^2 \sqsubseteq \cdots \sqsubseteq f_{b,c}^n \sqsubseteq \ldots$ (we can see that whenever $f_{b,c}^{n-1}$ is defined it agrees with $f_{b,c}^n$) and $f_{b,c}$ is monotonic ($f_{b,c}^k \sqsubseteq f_{b,c}^l \implies f_{b,c}(f_{b,c}^k) \sqsubseteq f_{b,c}(f_{b,c}^l)$).

For a given $X = x$, $f_{b,c}^{x+1} = f_{b,c}^{x+2} = \ldots$ . The fixed point is the lub of the whole sequence is therefore:

$$f_{b,c}^\infty = \bigsqcup_{n \ge 0} f_{b,c}^n(\bot) = \begin{cases} [X \mapsto 0, Y \mapsto y*(x!)] & \text{if } x > 0 \\ [X \mapsto x, Y \mapsto y] & \text{if } x \le 0 \end{cases}$$

By a similar analysis, it is not hard show that the denotation of $[\![\texttt{while true do skip}]\!]$ will be undefined.[8]

## C  Can algorithms, as implemented in CLRS-30, have an equilibrium?

In this appendix, we have copied over some algorithms implementations from CLRS-30[9]. Additionally, we have annotated how and when they follow the $\texttt{while } b \texttt{ do } c$ construct. Algorithms, that are

---

[8]omitted in this text – see Winskel [52]

[9]https://github.com/google-deepmind/clrs/tree/master/clrs/_src/

*not* necessarily solved via this construct (e.g. strongly connected components) were also included, so as to showcase if this would break.

```python
def bfs(A: _Array, s: int) -> _Out:
  chex.assert_rank(A, 2)
  probes = probing.initialize(specs.SPECS['bfs'])
  A_pos = np.arange(A.shape[0])
  probing.push(
      probes,
      specs.Stage.INPUT,
      next_probe={
          'pos': np.copy(A_pos) * 1.0 / A.shape[0],
          's': probing.mask_one(s, A.shape[0]),
          'A': np.copy(A),
          'adj': probing.graph(np.copy(A))
      })
  reach = np.zeros(A.shape[0])
  pi = np.arange(A.shape[0])
  reach[s] = 1
  # Initialisation code ends here

  # implemented as do-while, but do-whiles are essentially
  # c; while b do c
  while True:

    prev_reach = np.copy(reach)
    probing.push(
        probes,
        specs.Stage.HINT,
        next_probe={
            'reach_h': np.copy(prev_reach),
            'pi_h': np.copy(pi)
        })
    # command c: update reachability.
    for i in range(A.shape[0]):
      for j in range(A.shape[0]):
        if A[i, j] > 0 and prev_reach[i] == 1:
          if pi[j] == j and j != s:
            pi[j] = i
          reach[j] = 1

    if np.all(reach == prev_reach): # boolean condition b: has reachability vector
     changed
      break

  probing.push(probes, specs.Stage.OUTPUT, next_probe={'pi': np.copy(pi)})
  probing.finalize(probes)
  return pi, probes
```

Listing 1: BFS algorithm. Clearly implemented as while *b* do *c*.

```python
# The sampler
class FloydWarshallSampler(Sampler):
  """Sampler for all-pairs shortest paths."""

  def _sample_data(
      self,
      length: int,
      p: Tuple[float, ...] = (0.5,),
      low: float = 0., # never changed in the data generation
      high: float = 1., # never changed in the data generation
  ):
    graph = self._random_er_graph( # samples random ER graph with weights in [low;high)
        nb_nodes=length,
        p=self._rng.choice(p),
        directed=False,
        acyclic=False,
        weighted=True,
        low=low,
        high=high)
    return [graph]

# The implementation
def floyd_warshall(A: _Array) -> _Out:
  """Floyd-Warshall's all-pairs shortest paths (Floyd, 1962)."""

  chex.assert_rank(A, 2)
  probes = probing.initialize(specs.SPECS['floyd_warshall'])

  A_pos = np.arange(A.shape[0])
```

```python
30
31    probing.push(
32        probes,
33        specs.Stage.INPUT,
34        next_probe={
35            'pos': np.copy(A_pos) / A.shape[0],
36            'A': np.copy(A),
37            'adj': probing.graph(np.copy(A))
38        })
39
40    D = np.copy(A)
41    Pi = np.zeros((A.shape[0], A.shape[0]))
42    msk = probing.graph(np.copy(A))
43
44    for i in range(A.shape[0]):
45      for j in range(A.shape[0]):
46        Pi[i, j] = i
47
48    # Initialisation code ends here
49
50    # for loops are while loops
51    # ''for k in range(N)'' is equivalent to ''k:=0; while (i<N) do (c; k:=k+1)''
52    # NOTE #1 The NN, however, has to learn to increase the k, solely
53    # based on the 'pos' input feature; having a 'pred' feature,
54    # (as in string algorithms) might have been more appropriate
55
56    # NOTE #2 *Since the sampler (above) only samples positive edge weights*
57    # no negative-weight cycles can exist. Hence any reruns of the inner two
58    # for loops, after k iteraitons have passed will not change matrix D.
59
60    # boolean condition b: k iterations have passed
61    # (a necessary, *but not sufficient* condition is that D remains unchanged)
62    for k in range(A.shape[0]):
63      prev_D = np.copy(D)
64      prev_msk = np.copy(msk)
65
66      probing.push(
67          probes,
68          specs.Stage.HINT,
69          next_probe={
70              'Pi_h': np.copy(Pi),
71              'D': np.copy(prev_D),
72              'msk': np.copy(prev_msk),
73              'k': probing.mask_one(k, A.shape[0])
74          })
75
76      # command c: update D for intermediate vertex k
77      for i in range(A.shape[0]):
78        for j in range(A.shape[0]):
79          if prev_msk[i, k] > 0 and prev_msk[k, j] > 0:
80            if msk[i, j] == 0 or prev_D[i, k] + prev_D[k, j] < D[i, j]:
81              D[i, j] = prev_D[i, k] + prev_D[k, j]
82              Pi[i, j] = Pi[k, j]
83            else:
84              D[i, j] = prev_D[i, j]
85            msk[i, j] = 1
86
87    probing.push(probes, specs.Stage.OUTPUT, next_probe={'Pi': np.copy(Pi)})
88    probing.finalize(probes)
89
90    return Pi, probes
```

Listing 2: Floyd-Warshall algorithm and its sampler (above). Can also be viewed as while *b* do *c*, *in CLRS-30*.

```python
1  def strongly_connected_components(A: _Array) -> _Out:
2    """Kosaraju's strongly-connected components (Aho et al., 1974)."""
3
4    chex.assert_rank(A, 2)
5    probes = probing.initialize(
6        specs.SPECS['strongly_connected_components'])
7
8    A_pos = np.arange(A.shape[0])
9
10   probing.push(
11       probes,
12       specs.Stage.INPUT,
13       next_probe={
14         # < omitted for brevity >
15       })
```

```python
16
17    scc_id = np.arange(A.shape[0])
18    color = np.zeros(A.shape[0], dtype=np.int32)
19    d = np.zeros(A.shape[0])
20    f = np.zeros(A.shape[0])
21    s_prev = np.arange(A.shape[0])
22    time = 0
23    A_t = np.transpose(A)
24
25    # Initialisation code ends here
26
27    # boolean condition b: there are unvisited (color[s]=0) vertices
28    for s in range(A.shape[0]):
29      if color[s] == 0:
30        s_last = s
31        u = s
32        v = s
33        probing.push(
34            probes,
35            specs.Stage.HINT,
36            next_probe={
37              # < omitted for brevity >
38            })
39        # command c: grey them (color=1) and recursively visit descendants
40
41        # NOTE command c is another while b' do c' with a stack
42        while True: # b': stack is not empty
43          if color[u] == 0 or d[u] == 0.0:
44            time += 0.01
45            d[u] = time
46            color[u] = 1
47            probing.push(
48                probes,
49                specs.Stage.HINT,
50                next_probe={
51                  # < omitted for brevity >
52                })
53          for v in range(A.shape[0]): # c': add lowest id unvisited descendant of top-stack
      node to the top of the stack
54            if A[u, v] != 0:
55              if color[v] == 0:
56                color[v] = 1
57                s_prev[v] = s_last
58                s_last = v
59                probing.push(
60                    probes,
61                    specs.Stage.HINT,
62                    next_probe={
63                      # < omitted for brevity >
64                    })
65                break
66
67          if s_last == u: # no descending
68            color[u] = 2
69            time += 0.01
70            f[u] = time
71
72            probing.push(
73                probes,
74                specs.Stage.HINT,
75                next_probe={
76                  # < omitted for brevity >
77                })
78
79            if s_prev[u] == u: # and we are on top of the recursion
80              # although imaginary, in the implementation here,
81              # if a stack was used, it'd be empty in this if
82              # statement
83              assert s_prev[s_last] == s_last
84              break
85            pr = s_prev[s_last]
86            s_prev[s_last] = s_last
87            s_last = pr
88
89          u = s_last
90
91    color = np.zeros(A.shape[0], dtype=np.int32)
92    s_prev = np.arange(A.shape[0])
93
94    # boolean condition b'': there are unvisited (color[s]=0) vertices
95    # (order of visiting depends on finishing time;
```

```
96     #  see Introduction to algorithms , 4th edition , Chapter 20)
97     for s in np.argsort(-f):
98       if color[s] == 0:
99         s_last = s
100        u = s
101        v = s
102        probing.push(
103            probes ,
104            specs.Stage.HINT,
105            next_probe={
106              # < omitted for brevity >
107            })
108        # NOTE command c is another while b''' do c''' with a stack
109        while True: # b''': stack is not empty
110          scc_id[u] = s
111          if color[u] == 0 or d[u] == 0.0:
112            time += 0.01
113            d[u] = time
114            color[u] = 1
115            probing.push(
116                probes ,
117                specs.Stage.HINT,
118                next_probe={
119                  # < omitted for brevity >
120                })
121          for v in range(A.shape[0]): # c''': add lowest id unvisited descendant of top-
     stack node to the top of the stack
122            if A_t[u, v] != 0:
123              if color[v] == 0:
124                color[v] = 1
125                s_prev[v] = s_last
126                s_last = v
127                probing.push(
128                    probes ,
129                    specs.Stage.HINT,
130                    next_probe={
131                      # < omitted for brevity >
132                    })
133                break

135          if s_last == u:
136            color[u] = 2
137            time += 0.01
138            f[u] = time

140            probing.push(
141                probes ,
142                specs.Stage.HINT,
143                next_probe={
144                  # < omitted for brevity >
145                })

147            if s_prev[u] == u: # same as before
148              assert s_prev[s_last] == s_last
149              break
150            pr = s_prev[s_last]
151            s_prev[s_last] = s_last
152            s_last = pr

154          u = s_last

156   probing.push(
157       probes ,
158       specs.Stage.OUTPUT ,
159       next_probe={'scc_id': np.copy(scc_id)},
160   )
161   probing.finalize(probes)

163   return scc_id , probes
```

Listing 3: Kosaraju's strongly connected components [40] algorithm. It is composed of *four* (two nested ones, sequenced one after the other) `while` $b$ `do` $c$ constructs.

# D   Differences to Ibarz et al. [29]

Our differences are mostly required by software engineering rather than research, hence they live here. Differences are:

- Different DL framework (Pytorch [36])

- Ibarz et al. [29] use an extra nonlinearity after the GNN step. We found this to be not necessary (there are plenty of nonlinearities at the message function) for the baseline and to be making the training of DEARs less stable so we removed it.

- Sorting-based algorithms use a Sinkhorn operator to force the output to be a permutation. However, this gave very negative logits for the predictions at initialisation, leading to runs starting from a very high loss and converging to poorer minima. We fixed this by adding an off-centred leaky ReLU activation with the kink point at (-6, -6) right after the Sinkhorn operator. After conversion of logits to outputs via softmax, our change is mathematically equivalent to saying that the probability for each other node to be predecessor should not drop below $10^{-6}$.

## E  Picking least fixed point

For a given batch fixed point finding continues until all instances in the batch converge and *at each step* the solver is stepped *on all* instances. For a given instance, when two $\mathbf{H}^{(t)}$ and $\mathbf{H}^{(t')}$ are under the threshold $\delta$, for some $t \leq t'$, the `torchdeq` library prefers the state that has the lower distance to next state. Consequently, out the returned fixed points only one is guaranteed to be least – the one that require the most steps. This not only misaligns with domain theory, but also had the practical effect that the neural models require more iterations to converge the more we train them. Thus, we changed the library to choose the first $\mathbf{H}(t)$ that passes the fixed point criteriaq.

## F  Alignment algorithm

Assume we have computed pairwise distance matrices between the states and those are stored in a $T \times T_{\mathcal{G}}$ distance matrix $\mathbf{D}$ with elements $d_{i,j}$. Ignoring the required alignment of the last states, we focus on aligning the rest of the states. This is done via standard dynamic programming algorithm. The dynamic programming state we define is as follows: $dp_{i,j}$ is the most optimal alignment for the first $i$ DEAR states and first $j$ NAR states, with $dp_{0,j} = 0$ (having leftover NAR states mean we skipped some, but we do not want to penalise that) and $dp_{i,0} = \infty$ (we want to align all DEAR states). We consider two recursive formulas, first one we use, the other we use when $T \leq T_{\mathcal{G}}$:

- when the $T > T_{\mathcal{G}}$, there are extra states. To avoid infinities we will allow for two DEAR states to align to a same state:

$$dp_{i,j} = \min \begin{cases} dp_{i-1,j} + d_{i,j} & \text{aligning DEAR state } i \text{ and NAR state } j, \text{ but allowing for} \\ & \text{previous states to align to it as well} \\ dp_{i,j-1} & \text{skipping alignment with state } j \end{cases}$$

(5)

- when the $T \leq T_{\mathcal{G}}$ we require that each DEAR state aligns to an unique NAR state:

$$dp_{i,j} = \min \begin{cases} dp_{i-1,j-1} + d_{i,j} & \text{aligning DEAR state } i \text{ and NAR state } j \\ dp_{i,j-1} & \text{skipping alignment with state } j \end{cases}$$

(6)

We have highlighted the difference to the above in purple.

The optimal alignment for the whole two sequence is stored in $dp_{T,T_{\mathcal{G}}}$. As both $\mathbf{H}^{(0)}$ and $\mathbf{H}^{(0)}_{\mathcal{G}}$ are concatenation of 0 vectors (due to how we initialise the latent state), their distance is always 0 and they will always align as required. To enforce alignment of the last state, we take the optimal value for the subsequences without last states $dp_{T-1,T_{\mathcal{G}}-1}$ and *always* (even when subsampling, see below) add the distances between the last states to the loss function.

As the above will always penalise longer DEQ trajectories, we divide $dp_{T-1,T_{\mathcal{G}}-1}$ by $T-1$ before including it in the loss function. Lastly, to allow for "intermediate" states (ones not necessarily matching a GNN state) to exist, we subsample randomly, without replacement, $T' = \max(\lfloor \frac{T-1}{2} \rfloor, 1)$ DEAR states and apply the dynamic programming algorithm with the subsampled sequence.

# G    Training loss: NAR vs DEAR

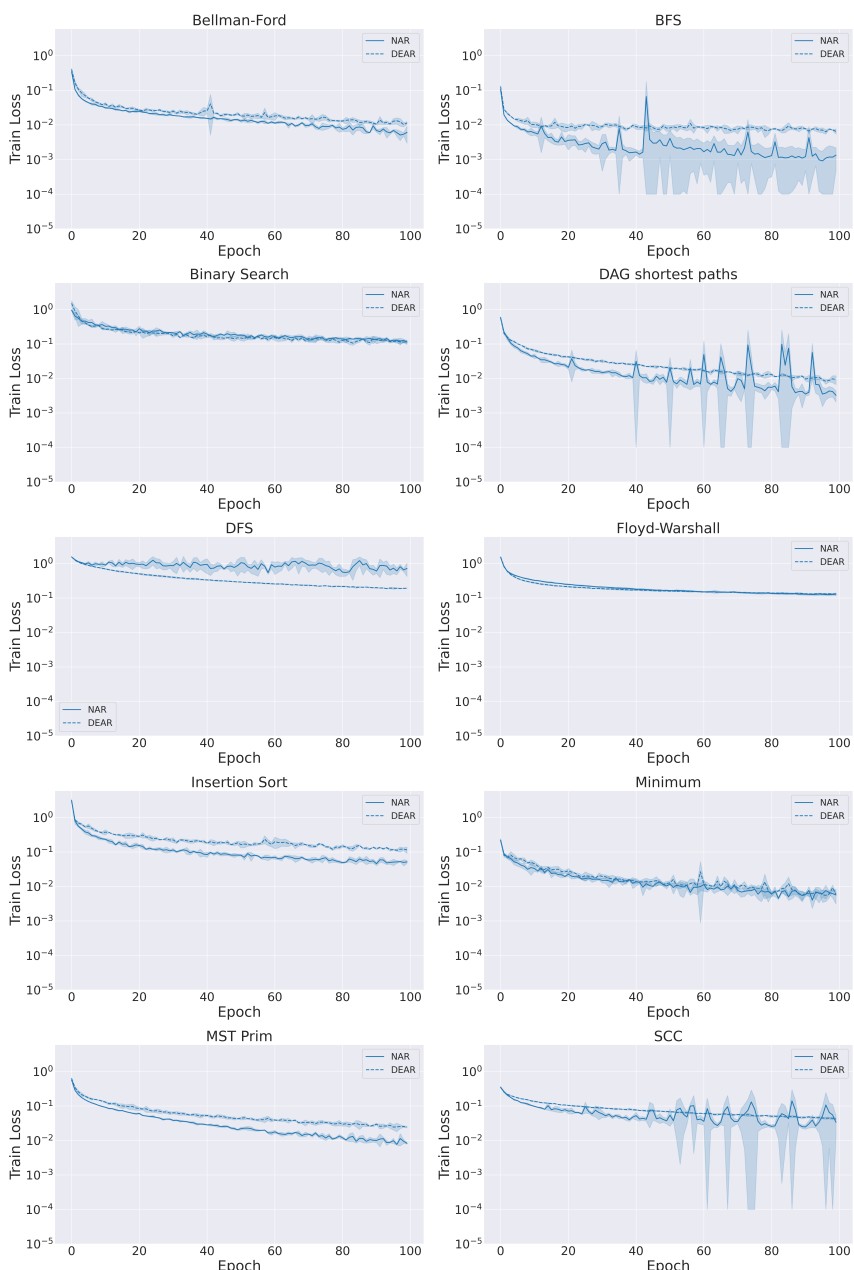

Figure 6: Side-by-side comparison of NAR vs DEAR. DEAR training loss is always within 1 order of magnitude of NAR. Note the log scale.

Table 5: Fixing anomalies with CLRS-30's binary search further increases our overall score making our approach very competitive to Triplet-MPNN. Notation taken from Table 1.

| Algorithm | NAR$^\blacklozenge$ | NAR$^\blacklozenge$ (Triplet-MPNN) | NAR$^\lozenge$ (LT) | DEAR (ours) |
|---|---|---|---|---|
| **Search** (Parallel) | $95.67\% \pm 0.58$ | $93.33\% \pm 0.58$ | $93.33\% \pm 3.05$ | $85.67\% \pm 0.58$ |
| **New Overall** | $71.52\%$ | $77.58\%$ | $70.97\%$ | $78.38\%$ |

## H  Binary search anomalies

In the CLRS-30 implementation of binary search, we aim to find the place to insert the target value `x` in a sorted array `A`. Thus we need to point to the graph node that holds the smallest value in the array `A`, which is greater than `x`. However, if `x>max (A)`, the answer is a pointer to the last value of the array, which by the convention used by CLRS-30 means we would be inserting `x` at the wrong place. In other words, the answer to `A=[0.1, 0.2, 0.3] x=0.25` and `A=[0.1, 0.2, 0.3] x=0.35` is the same – insert `x` to the left of 0.3. This contributed some noise, so we fixed the sampler to always give `x` within `[0, max(A))`. The other changes were to explicitly use `graph & pointer` instead of `node & mask_one` as the location & datatype of the pointer to the position in the array, also done by Engelmayer et al. [16]. Similarly to them, we also add an additional supervision signal, but at the output level rather than the hint level – teaching the models to predict which array elements are smaller than `x` (binary `mask` type).

We reran the new, parallel version of search, reporting results in Table 5. Our model still falls short of the baselines, but the 26% increase in accuracy is large enough to give a slight overall advantage to DEAR over the Triplet-MPNN model. We do note, however, that the task of searching is mostly numerical (comparison between floating point numbers), resulting in DEAR overfitting a lot – recall that train accuracy was 95% even for the original (binary) search. We verified that if the training data is increased $3 \times -5\times$, test accuracy becomes comparable to other models, regardless of which version is used.

# I  Training loss: DEAR vs DEAR w/ CGP

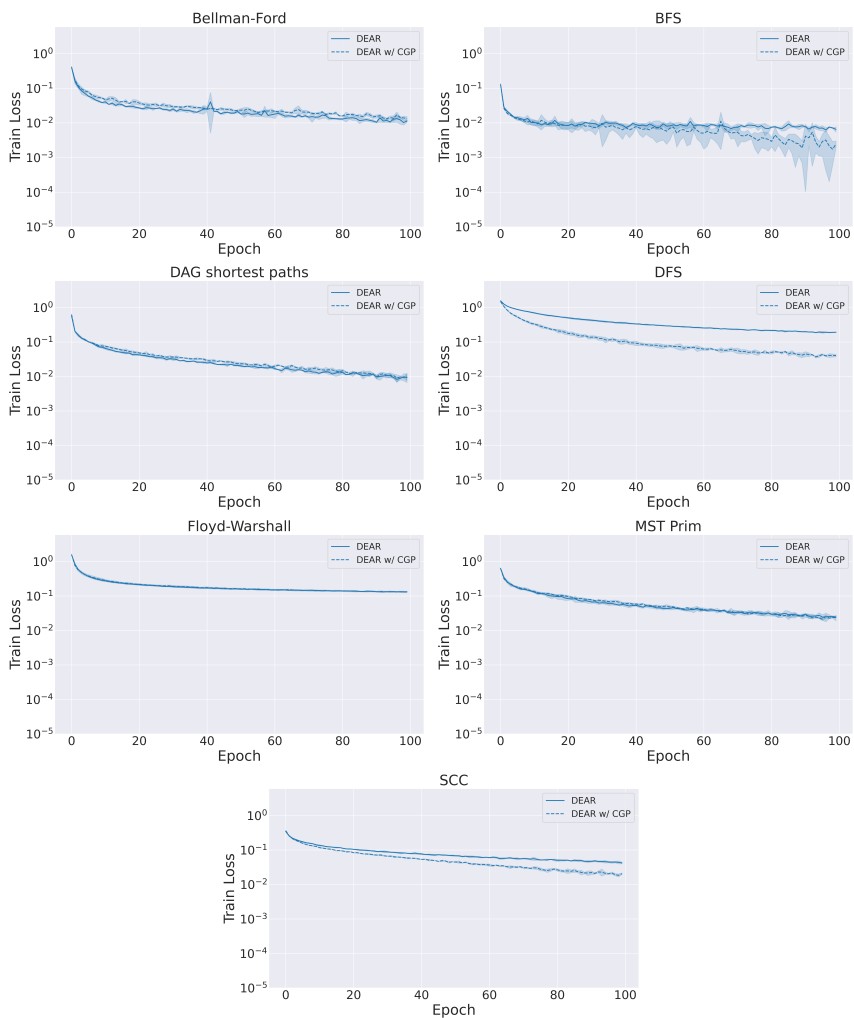

Figure 7: Effect of using Cayley Graph propagation on the train loss.

# J   Alignment gives closer convergence

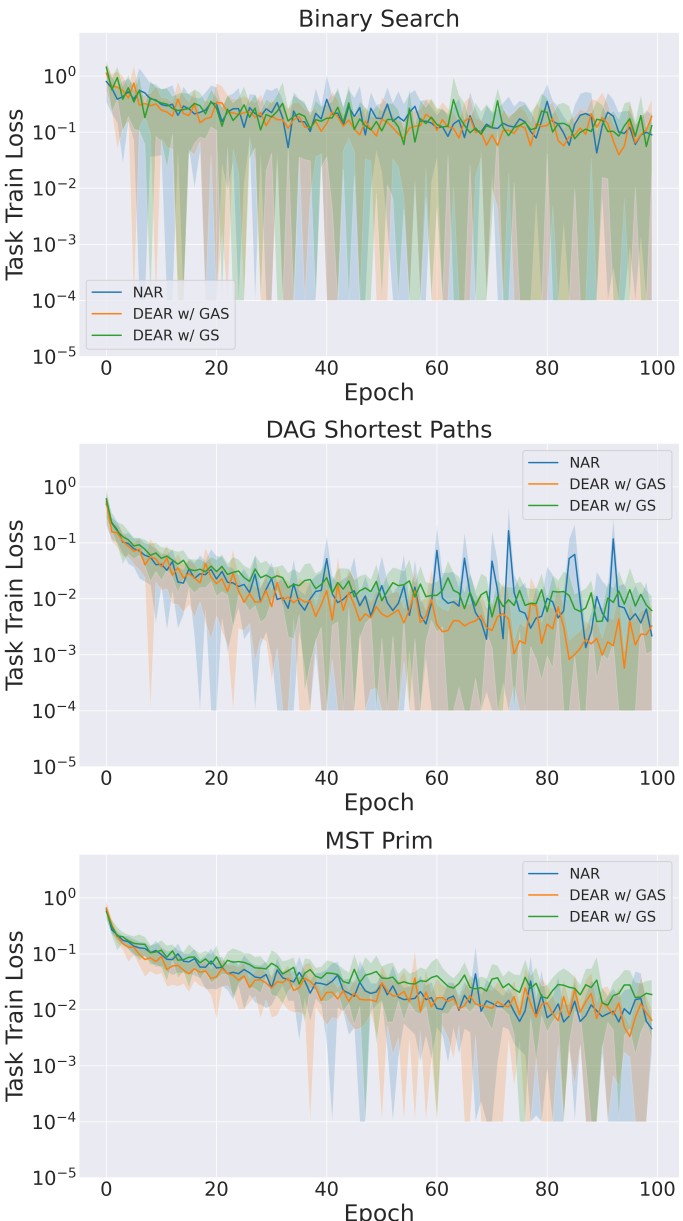

Figure 8: Alignment (orange) leads to lower task train loss compared to no aligment, but using stochasticity and GRANOLA (green).

