# OpenReview forum: "Deep Equilibrium Algorithmic Reasoning"
_NeurIPS.cc/2024/Conference — NeurIPS 2024 poster_

### Official Review · Reviewer_E2D4 · 2024-06-14

**Soundness:** 3
**Presentation:** 4
**Contribution:** 3
**Rating:** 6
**Confidence:** 1

**Summary:**

This work builds deep equilibrium graph neural networks for algorithmic reasoning.  The paper tests their models on a variety of algorithm problems and finds mixed results with some positive and encouraging observations.  One focus of the work is on speeding up NARs, and the paper also proposes regularizers to boost performance. To be honest, I am not an algorithms person, and a lot of ideas in the paper are foreign to me, so I am not confident about my review.

**Strengths:**

The paper is well-written, polished, and clear.  In general, the idea of learning algorithms with neural networks has the potential for practical value, although it seems to me that this paper focuses on emulating existing known algorithms and not learning new ones.  Also, the experiments show encouraging results.

**Weaknesses:**

The results seem mixed and not entirely positive.

In my own experiments with DEQs, I’ve found that often they overfit to the solver used during training.  Other solvers can find fixed points, but those fixed points don’t always map on to solutions.  Whether or not the model actually has unique fixed points corresponding to solutions may be worth studying.

The paper mentions another paper on DEQs for algorithmic reasoning (“Deep Equilibrium Models For Algorithmic Reasoning”), but does not discuss it in detail.  This other paper has a nearly identical title and probably demands a more careful discussion and contextualization.  As is, it would be easy for a reader to think that this paper is the first one to think about DEQs for algorithmic reasoning.

It might be worth discussing the work on non-GNN recurrent networks for learning algorithms.  For example, I have found that a model from "End-to-end Algorithm Synthesis with Recurrent Networks: Logical Extrapolation Without Overthinking" actually behaves similar to DEQs in practice.

**Questions:**

N/A

**Limitations:**

The authors adequately discuss limitations throughout the paper.

---

> ### Author Rebuttal · Authors · 2024-08-06
>
> Many thanks to the reviewer for their thoughtful review and for finding our paper’s presentation excellent. Allow us to address both the comments and questions you have raised.
>
> *The results seem mixed and not entirely positive*
>
> The effectiveness of DEAR may be misinterpreted due to the mixture of results for each algorithm. The varied algorithms found in CLRS-30 means that some models are expected to perform better on some tasks compared to others. To highlight the strengths of our model, we have added an average across all algorithms (overall row) found in Table 1 in the rebuttal document.
>
> The main comparison of DEAR is against the baseline NAR, which has an overall performance increase of 4% (Table 1). The NAR model and other non-equilibrium models have access to exact number of steps at both train and test time. The processor (a recurrent GNN) is unrolled for the given number of steps (may differ between samples). As shown in Table 1 for NAR (HCS), the model is susceptible to step change. For certain algorithms 64 steps may be just right, but for others it might be more or less, hardcoding the number of steps makes the performance 15% worse than DEAR.
>
> A fairer comparison is the NAR (LT) model (Table 1; rebuttal PDF), which has a trained architecture to decide the number of steps during test time; we outperform by 5%. To reiterate, this model is still trained on the exact number of steps.
>
> To further motivate the strengths of our model we opted to include the state-of-the-art processor architecture of Triplet-MPNN. Commendably, our model performs 2% worse than a model that has many other improvements besides equilibrium. In Table 1 there are two versions of Triplet-MPNN, the latter includes causality based regularisation. We had to copy results from related work (Bevilacqua et al., 2023) as we do not have access to their implementation. Triplet-MPNN is a very strong baseline to compare against, as each categorically-inspired GNN layer considers interactions between each edge and nodes that are not necessarily connected to the edge. This edge-node interaction, of course, comes at the expense of having to materialise O(V^3) messages. Notably, the DEAR approach can be used in conjunction with Triplet-MPNN (Table 5; rebuttal PDF), giving a state-of-the-art overall performance (Table 5; rebuttal PDF).
>
> Finally, we investigated binary search as it was an anomaly for DEAR’s performance. We found several issues for binary search in CLRS-30: sampling process, calculation of ground truths and misspecification of output type. As a result, we ran new baselines of the search algorithm (Table 2; rebuttal PDF). With this change, the new overall for DEAR is 7% higher than the baseline NAR, and comparable to both Triplet-MPNN variants.
>
> *In my own experiments with DEQs, I’ve found that often they overfit to the solver used during training…*
>
> We thank the reviewer for providing this observation. We have investigated this, in the context of the search algorithm. We observed improved performances indeed (by ~1%) but we found the increase not as substantial compared to other ablations presented in the rebuttal PDF. We integrate this result and other algorithms for the final version of the paper. Nonetheless, this could pose an interesting setting for future exploration.
>
> *The paper mentions another paper on DEQs for algorithmic reasoning …, but does not discuss it in detail…*
>
> The concurrent work we highlight in our paper does indeed follow a similar research direction and was submitted to a peer-reviewed conference earlier this year, hence the citation. However it is not a paper, but rather a blogpost, and as a result we strongly recommend the reviewer to treat it as such. Of course blogposts are essential for progressing science by publicly exchanging ideas, but pragmatically the timeline from idea creation to a paper publication differs greatly.
>
> The key differences between our paper and the blogpost are in how we approach using equilibrium points in NAR. We do not claim we were the first to conjecture the existence of this connection (we accredit this to the blogpost). However, we do claim:
> * We thoroughly formalise the DEQ-NAR connection
> * Following the formalisation we build the first robust equilibrium algorithmic reasoner; The model from the blogpost performs terribly on the simplest task of BFS.
> * Our model outperforms non-equilibrium baselines and is competitive to a model using a more expressive GNN
> * We show our model is also efficient
>
> *It might be worth discussing the work on non-GNN recurrent networks for learning algorithms...*
>
> Ideas from this paper are already implemented in our baselines as well as our equilibrium reasoners (eq. 5, L175). U and E are embeddings of input node/edge features, something given to the DEAR model at each step, which is the recall feature. A similar thing to an incremental progress training algorithm has also been seen in NAR, where trajectories are chunked into independent segments, but we did not find it necessary for the good performance in NAR.
>
> Once again we thank the reviewer for their insightful comments and excellent questions, especially in regards to their insights about DEQs that present interesting avenues of future work. Consequently, even though you state you’re not an “algorithm person”, we strongly respect your viewpoints and intuition and we believe your confidence score can be improved; a 3 seems like the minimum as per reviewer guidelines.
>
> Finally, we hope that our rebuttal addresses the empirical strengths of the DEAR model via the updated results and the differences between the blogpost and paper, such that it may convince you to increase your score. We are of course happy to further engage with the reviewer for any remaining doubt.

---

> > ### Comment · Reviewer_E2D4 · 2024-08-12
> > **Thanks for your response**
> >
> > Thanks for your response.  I keep my review.

---

> ### Author Response · Authors · 2024-08-12
>
> Thank you for taking the time to read our rebuttal. We believe we have addressed all your concerns, therefore could we kindly ask if there is any particular reason why you have not increased your score or confidence?  We found your review insightful and would be happy to listen to further comments. We are available for any additional concerns or questions.

---

### Official Review · Reviewer_tzFP · 2024-06-14

**Soundness:** 2
**Presentation:** 2
**Contribution:** 2
**Rating:** 6
**Confidence:** 1

**Summary:**

This paper proposes to solve the neural algorithmic reasoning by attacking the equilibrium solutions directly, without leveraging the recurrent structures which imitate the iterations in algorithms. They proposed deep equilibrium algorithmic reasoner (DEAR), and compare it with baselines including NAR (w/ and w/o Triplet-MPNN).Their training dynamics is shown to be more stable than baselines, and inference time is smaller than than baselines, although the accuracy is sometimes worse than baselines.

**Strengths:**

* The motivation is clear.
* Experiments are thorough.

**Weaknesses:**

* It is not immediately clear what is the takeaway from the experiments. My understanding is that efficiency is the key selling point of the new algorithm. However, there is only one table for efficiency in Appendix G.
* Section 4 is notation heavy and hard to understand.
* I fully appreciate authors' honesty to report the results (Table 1) which are not to their advantage, but the inferior accuracy seems to limit the usefulness of the method.

**Questions:**

* I'm not an expert so I have trouble understanding Section 4. Is it possible to make it clearer how does it connect to the proposed algorithm?
* Eq. (4) seems still a recurrent structure? I was thinking the point of the new method is to get rid of recurrent structures? But looks like from Figure 1, the point is to reduce the number of recursive steps. Is this correct?

**Limitations:**

The authors adequately addressed the limitations.

---

> ### Author Rebuttal · Authors · 2024-08-06
>
> Many thanks to the reviewer for their thoughtful review, and allow us to address both the comments and questions you have raised.
>
> *It is not immediately clear what is the takeaway from the experiments. My understanding is that efficiency is the key selling point of the new algorithm…*
>
> The key selling point is that finding the equilibrium is a proper way to decide termination during training and inference time. Equilibrium also acts as an additional architectural bias, giving further boost in accuracy (Table 1; rebuttal PDF). Fixing the number of steps at test time has detrimental effects. Using a dedicated architecture to learn when to terminate the algorithm also falls short of our approach.
>
> *Section 4 is notation heavy and hard to understand.*
>
> We thank the reviewer for raising this point as we understand that these topics may be unfamiliar to readers from the deep learning domain.
>
> As a result, we improved this section to provide a more readable and comprehensive understanding of domain theory. Upon taking your advice, we also added an appendix solely dedicated to the IMP language and added more details regarding the denotation of a while loop in the appendix.
>
> Unfortunately, the NeurIPS rules prevent us from uploading a revised version of the paper. Some of the key improvements to provide yourself the confidence that we have addressed this concern are:
> * Analogies with the elements of popular programming languages, such as C
> * More details on the definition of States
> * Expanded the definition of Commands with more examples
> * More details on the utility of Domain Theory in our setting
> * Further motivation for our theoretical excursion
>
> The goal of our theoretical analysis is to show that there is a well formalised relationship between equilibrium models and algorithms.
>
> *I fully appreciate authors' honesty to report the results (Table 1) which are not to their advantage…*
>
> The effectiveness of DEAR may be misinterpreted due to the mixture of results for each algorithm. The varied algorithms found in CLRS-30 means that some models are expected to perform better on some tasks compared to others. Thus to highlight the strengths of our model, we have added an average across all algorithms (overall row) found in Table 1 in the rebuttal document.
>
> The main comparison of DEAR is against the baseline NAR, which has an overall performance increase of 4% (Table 1). The NAR model and other non-equilibrium models have access to exact number of steps at both train and test time. The processor (a recurrent GNN) is unrolled for the given number of steps (may differ between samples). As shown in Table 1 for NAR (HCS), the model is susceptible to step change. For certain algorithms 64 steps may be just right, but for others it might be more or less, hardcoding the number of steps makes the performance 15% worse than DEAR.
>
> A fairer comparison is the NAR (LT) model (Table 1; rebuttal PDF), which has a trained architecture to decide the number of steps during test time; we outperform by 5%. To reiterate this model is still trained on the exact number of steps.
>
> To further motivate the strengths of our model we opted to include the state-of-the-art processor architecture of Triplet-MPNN. Commendably, our model performs 2% worse than a model that has many other improvements besides equilibrium. In Table 1 there are two versions of Triplet-MPNN, the latter includes causality based regularisation. We had to copy results from related work (Bevilacqua et al., 2023) as we do not have access to their implementation. Triplet-MPNN is a very strong baseline to compare against, as each categorically-inspired GNN layer considers interactions between each edge and nodes that are not necessarily connected to the edge. This edge-node interaction, of course, comes at the expense of having to materialise O(V^3) messages. Notably, the DEAR approach can be used in conjunction with Triplet-MPNN (Table 5; rebuttal PDF), giving a state-of-the-art overall performance (Table 5; rebuttal PDF).
>
> Finally, we investigated binary search as it was an anomaly for DEAR’s performance. We found several issues for binary search in CLRS-30: sampling process, calculation of ground truths and misspecification of output type. As a result, we ran new baselines of the search algorithm (Table 2; rebuttal PDF). With this change, the new overall for DEAR is 7% higher than the baseline NAR, and comparable to both Triplet-MPNN variants.
>
> *…Is it possible to make it clearer how does it connect to the proposed algorithm?*
>
> Section 4 serves as a formal motivation for using neural equilibrium models when learning to simulate algorithms. What we propose is not an algorithm per se, but rather we propose a new equilibrium-based way to decide termination of algorithm simulation both during training and inference. The most elegant way (in our opinion) to formalise this concept is through denotational semantics in comparison with other approaches, such as coalgebras (category theory concept).
>
> In our revised version we have emphasised the connection between DEAR and denotational semantics, especially the paragraph “Finding the fixed point” (section 5, L178 of our paper draft), by referring specifically to the concepts defined in Section 4.
>
> *Eq. (4) seems still a recurrent structure? ... new method is to get rid of recurrent structures? But ..., the point is to reduce the number of recursive steps. Is this correct?*
>
> The point is to find a robust way to decide termination in neural algorithmic reasoning that doesn’t greatly compromise accuracy or efficiency.
>
> Once again we thank the reviewer for their insightful comments and excellent questions. We hope our reply addresses all concerns and questions, such that it may convince you to increase your score. We are of course happy to further engage with the reviewer for any remaining doubt.

---

> > ### Comment · Reviewer_tzFP · 2024-08-12
> >
> > I want to thank the authors for nicely addressing my concerns. Therefore, I'm raising my score.

---

> ### Author Response · Authors · 2024-08-12
>
> We would like to sincerely thank you again for the insightful feedback from your rebuttal, and for additionally raising your score. We truly believe you helped strengthen our paper. Please let us know if you have any further suggestions to help us improve our work.

---

### Official Review · Reviewer_fLfp · 2024-07-01

**Soundness:** 2
**Presentation:** 2
**Contribution:** 2
**Rating:** 5
**Confidence:** 4

**Summary:**

This paper explores a novel approach to NAR using GNNs. Traditional NAR models typically use a recurrent architecture where each iteration of the GNN corresponds to an iteration of the algorithm being learned. Instead, this paper proposes that since many algorithms reach an equilibrium state where further iterations do not alter the outcome, it is possible to directly solve for this equilibrium. By training neural networks to find the equilibrium point, the authors aim to improve the alignment between GNNs and classical algorithms, potentially enhancing both the accuracy and speed of NAR models. Empirical evidence from the CLRS-30 benchmark supports the viability of this equilibrium-based approach.

**Strengths:**

1. The proposed method is clearly presented.

2. The performance of the proposed method on benchmark datasets is good.

**Weaknesses:**

1. The transition from denotational semantics to the proposed architecture could be further clarified to improve the overall understanding of the method. (Question 1)

2. It is not entirely clear whether the comparison with the baseline is accurate, and further justification or explanation may be needed. (Question 2)

3. The authors might consider revising the presentation of their contributions to more effectively convey the significance and novelty of the paper to the audience. (Question 3)

4. Including real-world experiments in the paper could substantially strengthen the contributions and further demonstrate the practical applicability of the proposed method. (Question 4, 5)

**Questions:**

1. From reviewer's understanding, denotational semantics primarily motivate the equilibrium methods.
- The connection between the proposed architecture (PGN with gating) and denotational semantics is not clear.
- The motivation for equilibrium methods does not seem to require denotational semantics, as graph algorithms such as Bellman-Ford inherently have equilibrium points.

Could the authors provide further clarification on this aspect?

2. It appears that the data for the experiments was generated by the authors rather than from CLRS-30, as they mention "For each algorithm, we generate" in Line 219. However, the baseline scores are identical to those reported in previous papers. To ensure a fair comparison, the authors should consider re-running the baselines on their own dataset.

3. From the reviewer's personal perspective, the introduction and contributions sections may not be well-suited for an academic paper in their current form. The investigated problem could benefit from better motivation, and the contributions might not fully capture the paper's key theoretical and empirical results.

4. The training and test graph sizes used in the experiments seem to be relatively small, although consistent with CLRS-30. The reviewer recommends training on larger graphs (e.g., 50 nodes) and testing on even larger graphs (e.g., 300 nodes) to better demonstrate the performance gains of the proposed methods.

5. The experiments presented in the paper are limited to synthetic graph datasets. The reviewer suggests that demonstrating the performance gains on real tasks (e.g., physical systems [1] or mathematical problems [2]) would enhance the contributions of this paper.

6. The authors mention that DEQ models can be difficult to train, which might be due to the instability of DEQ's gradient. The reviewer recommends considering the use of [3] for training the models, as it provides a more stable approach.

[1] Battaglia, Peter, et al. "Interaction networks for learning about objects, relations and physics." NeurIPS 2016.

[2] Lample, Guillaume, and François Charton. "Deep learning for symbolic mathematics." ICLR 2020.

[3] Fung, Samy Wu, et al. "Jfb: Jacobian-free backpropagation for implicit networks." AAAI 2022.

**Limitations:**

Yes

---

> ### Author Rebuttal · Authors · 2024-08-06
>
> Many thanks to the reviewer for their thoughtful review and directly linking the weaknesses with their questions. We address all the questions below.
>
> *The connection between the proposed architecture (PGN with gating) and denotational semantics is not clear.*
>
> Section 4 serves as a formal motivation for using neural equilibrium models when learning to simulate algorithms. We propose not an architecture (PGN was invented by Veličković et al. 2020), but a new way to decide termination of algorithm simulation both during training and inference. We have now emphasised the connection between DEAR and denotational semantics, especially the paragraph “Finding the fixed point” (section 5, L178 of our paper draft), by referring specifically to the concepts defined in Section 4.
>
> Lastly, our proposed model uses PGN due to being a lightweight and well-performant NAR architecture, which serves as the ideal baseline. However, DEAR is architecture agnostic and it can work with Triplet-MPNN (Table 5; rebuttal PDF).
>
> *The motivation… does not seem to require denotational semantics...*
>
> In our experience, we have observed sometimes intuition can mislead deep learning. (cf. “Understanding deep learning requires rethinking generalisation” by Zhang et al.) Hence, we decided to formally motivate the paper using denotational semantics. This inspired the alignment scheme proposed and the decision to pick the least fixed point if more than one fixed point exists.
>
> *It appears that the data for the experiments was generated by the authors … To ensure a fair comparison, the authors should consider re-running the baselines on their own dataset.*
>
> For clarification, the results are generated with our data and code except for Triplet-MPNN with casual registration (Bevilacqua et al., 2023) and DEM (Xhonneux et al., 2024); the implementation for both are not public, hence we reported theirs.
>
> However, we have taken your advice and generated our own baseline for Triplet-MPNN (Table 1; rebuttal PDF) without casual registration. Commendably, our model performs only 2% worse than a model whose categorically inspired GNN layer considers interactions between each edge and nodes that are not necessarily connected to the edge. This edge-node interaction, of course, comes at the expense of having to materialise O(V^3) messages. The rebuttal PDF (further explained in the global rebuttal) also highlights the strengths of our models compared to our new experiments. DEAR is 4% better than the baseline NAR model.
>
> *From the reviewer's personal perspective, the introduction and contributions sections may not be well-suited for an academic paper in their current form...*
>
> Unfortunately, the NeurIPS rules prevent us from uploading a revised version of the paper. We have rewritten our contributions to more clearly highlight the main outcomes of our paper, which are:
> * The DEQ-NAR connection is formally motivated
> * The first robust equilibrium algorithmic reasoner; the model from the DEM blogpost performs terribly on the simplest task of BFS.
> * A regularisation scheme to encourage alignment with algorithm execution traces when training deep equilibrium neural algorithmic reasoners
> * A comprehensive evaluation that shows DEAR is competitive to a model using a more expressive GNN
>
> *...The reviewer recommends training on larger graphs and  testing on even larger graphs to better demonstrate the performance gains ….*
>
> As highlighted by the reviewer, the current standard setting in literature is a training size of 16 nodes and a test size of 64 nodes. We appreciate the recommendation, however, given the tight time constraint, our limited hardware access and the requirement of training 10 algorithms, for 3 seeds, for all models (for a fair comparison) the total number of runs goes beyond 100. This provides an interesting setting for future work.
>
> Nevertheless, this comment inspired us to evaluate our currently trained models on larger instances (Table 3; rebuttal PDF); 128 nodes (8x), 256 nodes (16x) and 512 nodes (32x). The results highlight that besides certain algorithms (search; see global rebuttal) we are extremely competitive to the baseline. This is offset by the fact that our model is much faster (Table 4; rebuttal PDF; we never exceed 0.5s/sample at any scale) and speedups sometimes exceed 25x-40x.
>
> *The experiments presented in the paper are limited to synthetic graph datasets. The reviewer suggests that demonstrating the performance gains on real tasks ...*
>
> Our experimental procedure follows the standard in the NAR literature. However, the focus of this paper is presenting a different approach for unrolling an algorithm execution which can serve as a new foundational model in NAR Consequently, even though we agree that it is interesting to test these reasoners in real-world scenarios (Numeroso et al., 2023), this is out-of-scope for this paper and would require work that does not align with the NAR standards, our goals and experiments.
>
> *The authors mention that DEQ models can be difficult to train, … The reviewer recommends considering… [3]...*
>
> We thank the reviewer for the provided reference. However, in this work, we never mention that DEARs are difficult to train; they converge to a slightly larger final training loss, but we found the training overall stable. We kindly ask the reviewer to point us towards any confusing paragraphs and we will add a reference to the mentioned paper as it can be useful for future readers of our work.
>
> Once again we thank the reviewer for their insightful comments and excellent questions. We hope our reply addresses all concerns and questions, such that it may convince you to increase your score. We are of course happy to further engage with the reviewer for any remaining doubt.

---

> ### Comment · Reviewer_fLfp · 2024-08-08
>
> I appreciate the authors' efforts to address my concerns, which has led me to increase my evaluation to a score of 5. The size generalization performance of DEQ is particularly noteworthy, aligning with the out-of-distribution generalization capabilities demonstrated in previous DEQ studies.
>
> Concerning the first point, the term 'formal motivation' used by the authors in reference to denotational semantics has left me a bit perplexed. While I recognize the attempt to provide a theoretical grounding, it appears to me more as an intuition rather than leading to a rigorous formal derivation that substantiates the utility of DEQ in graph reasoning. Moreover, the connection between denotational semantics and graph reasoning problems seems tenuous, with the only clear link being the capability to solve such problems through programming.
>
> As for the second point, **I have reservations about the rigorousness of reusing scores from a previous paper when the test dataset has changed**, but I am not sure about it.

---

> > ### Author Response · Authors · 2024-08-09
> >
> > We thank the reviewer for promptly replying to our rebuttal and we’re glad we were able to address some of your concerns.
> >
> > *[..] the term 'formal motivation' used by the authors in reference to denotational semantics has left me a bit perplexed. [..] the connection between denotational semantics and graph reasoning problems seems tenuous [..]*
> >
> > We appreciate the comment from the reviewer, and we would like to clarify that we do not aim at providing a connection between "denotational semantics and graph reasoning problems", but rather to provide a connection between finding the fixed point of a function and executing an algorithm to termination. This then provides an analogy between finding the fixed point of a neural algorithmic reasoner and executing an algorithm to termination, which is the base motivation of our proposed method. We will change our language in our paper to remove any ambiguities and make clear that we use the mathematical background as a strong motivating factor for our method.
> >
> > *[..] I have reservations about the ethical implications of reusing scores from a previous paper [..]*
> > We understand the comment from the reviewer and we would like to add a comment, as perhaps the words used should be related to scientific rigour, rather than ethical concerns.
> >
> > Reporting results from other papers, providing proper citation and mention (we agree that it would be extremely unethical without this), is a common practice in literature, especially when the code for some models is not made public. We make sure to use the **same data generation code** used by all considered papers in the literature. For CLRS there isn't a predefined dataset, but rather everyone uses the same data-generation library, ensuring the same data distribution.
> >
> > We note however, that there is some [official downloading code](https://github.com/google-deepmind/clrs/blob/d1c2ad7af8437c7536cf329d9cef8fdf93184d9d/clrs/examples/run.py#L152) for test data generated with CLRS. All algorithms are reported in the results below except for SCC. This algorithm was skipped as pointers in the downloaded dataset were not in the edge-set. PyTorch Geometric does not support this feature and this difference will be mentioned in Appendix B within our paper.
> >
> > We report results with DEAR on all remaining algorithms. They are consistent with our data and even improve on DSP, Bellman-F. and Floyd-W..
> >
> > | Algorithm          | Mean    | Std Dev  |
> > |--------------------------|---------|----------|
> > | Bellman-F.       | 98.59%  | 0.34   |
> > | Floyd-W.       | 63.92%  | 0.04   |
> > | DSP           | 92.95%  | 1.38   |
> > | MST Prim            | 89.33%  | 1.09   |
> > | BFS                 | 99.65%  | 0.15   |
> > | DFS                  | 38.27%  | 0.82   |
> > | Search (Binary)       | 74.00%    | 11.5    |
> > | Minimum            | 99.33%  | 1.15   |
> > | Sort (Ins)    | 85.48%  | 6.90   |
> >
> >
> > We thank the reviewer again for all the insightful comments, and remain available for further clarifications if any doubts remain.

---

> ### Comment · Reviewer_fLfp · 2024-08-09
>
> Although the performance of the proposed method is superior, I am not convinced that the motivation outlined by the authors differs significantly from any other theoretically inspired motivations. Therefore, I will maintain my score as a borderline accept.

---

### Official Review · Reviewer_1Bds · 2024-07-13

**Soundness:** 3
**Presentation:** 2
**Contribution:** 3
**Rating:** 6
**Confidence:** 3

**Summary:**

This paper proposes Deep Equilibrium Algorithmic Reasoner (DEAR) which uses a deep equilibrium model (DEQ) to solve algorithmic tasks in CLRS30. The paper first introduces denotational semantics, which can be used to denote programs. It also provides an overview of Domain theory, and uses it to show that algorithms have fixed points. The paper then trains pointer graph network on algorithms from CLRS30 as a DEQ — fixed point solving is done with Anderson solver. The paper also discusses different methods (including failure cases) that were attempted to further improve performance of DEAR — use of Cayley Graph Propagation and alignment loss. DEAR improves performance over NAR (Neural Algorithmic Reasoner, the preivous state-of-the-art) on many algorithms.

**Strengths:**

1. The premise of this paper makes perfect sense. Message passing in GNNs is known to converge to an equilibrium. Therefore, it is obvious to combine NAR with DEQs.
2. DEAR improves OOD generalization of algorithms such as Floyd-Warshall, DFS, SCC, Sorting, and performs comparably to NAR (previous state-of-the-art) on algorithms like Breadth-first search, Minimum.
3. DEAR improves inference speed of many algorithms as shown in Appendix G.

Overall, this work is quite novel. While there are some prior works (See Anil et al. 2022) that have applied DEQs on very small GNNs and simple algorithmic tasks, this paper picks up a relatively difficult problem of solving standard algorithms. This hasn't been explored before  with DEQ based architecture.

**Weaknesses:**

1. DEAR hurts performance on some algorithms like Binary Search (significant drop in performance), DSP, and MST Prime, and the reasoning is unclear.
2. The authors have done an excellent job at explaining literature that many readers might not be familiar with. However, the readability of paper can be improved further if the authors provide more background on domain theory for those who are not familiar with it. It will also help if some discussion is added in Appendix A. I couldn’t understand it even after attempting to read it multiple times.

**Questions:**

1. Why use a partial function in equation 3?
2. There is some prior work [1] which indicates that more steps to find equilibrium points help with better OOD generalization. From my understanding, DEAR penalizes longer trajectories (due to misalignment with domain theory). Is it possible to get improved performance if we ignore the potential conflict with domain theory?
 [1] Anil, Cem, et al. "Path independent equilibrium models can better exploit test-time computation." Advances in Neural Information Processing Systems 35 (2022): 7796-7809.
3. What loss objective is used to train DEAR (e.g. cross entropy)? I understand that there are auxiliary losses such as alignment loss in Lines 277-298.

**Limitations:**

The limitations have been discussed.

---

> ### Author Rebuttal · Authors · 2024-08-06
>
> Many thanks to the reviewer for their thoughtful review, and allow us to address both the comments and questions you have raised.
>
> *DEAR hurts performance on some algorithms like Binary Search… and the reasoning is unclear.*
>
> Firstly, we would like to emphasise that the varied algorithms found in CLRS-30 means that some models are expected to perform better on some tasks compared to others. Thus to highlight the strengths of our model, we have added an average across all algorithms (overall row) found in Table 1 in the rebuttal document.
>
> As stated by the reviewer, the performance of binary search is an anomaly for DEAR’s performance, therefore we chose to investigate this algorithm further. We found several issues for binary search in CLRS-30 (Veličković et al. 2022): sampling process, calculation of ground truths, and misspecification of output type. Consequently, we ran new baselines of the search algorithm (Table 2; rebuttal PDF). With this change, the new overall for DEAR is 7% higher than the baseline NAR, and comparable to both Triplet-MPNN variants. Additionally, our decreased performance on binary search is due to overfitting. Our investigation shows that this is a data-hungry algorithm. To verify this claim, we trained 1 seed with three times larger training data (same number of epochs) and confirmed we got close to perfect performance.
>
> To provide further clarification of the strength of DEAR we will explain the empirical results found in the rebuttal PDF. The main comparison of DEAR is against the baseline NAR, which has an overall performance increase of 4% (Table 1). The NAR model and other non-equilibrium models have access to exact number of steps at both train and test time. The processor (a recurrent GNN) is unrolled for the given number of steps (may differ between samples). As shown in Table 1 for NAR (HCS), the model is susceptible to step change. For certain algorithms 64 steps may be just right, but for others it might be more or less, hardcoding the number of steps makes the performance 15% worse than DEAR.
>
> A fairer comparison is the NAR (LT) model (Table 1; rebuttal PDF), which has a trained architecture to decide the number of steps during test time; we outperform it by 5% overall and with ~1 standard deviation on DSP/MST Prim. To reiterate this LT model is still trained on the exact number of steps, something never given to our model.
>
> Finally, the DEAR approach is independent of the GNN processor architecture, therefore it can be used in conjunction with the state-of-the-art Triplet-MPNN. In Table 5, we train DEAR with TripletMPNN on the subset of the algorithms Triplet-MPNN improves the most. The overall accuracy was increased by 4%, when using DEAR, suggesting that our approach is also architecture agnostic.
>
> *The authors have done an excellent job at explaining literature ... However, the readability of paper can be improved further if the authors provide more background on domain theory ... It will also help if some discussion is added in Appendix A...*
>
> We thank the reviewer for raising this point as we understand that these topics may be unfamiliar to readers from the deep learning domain.
>
> As a result, we improved this section to provide a more readable and comprehensive understanding of domain theory. Upon taking your advice, we also added an appendix solely dedicated to the IMP language and added more details regarding the denotation of a while loop in the appendix.
>
> Unfortunately, the NeurIPS rules prevent us from uploading a revised version of the paper. Some of the key improvements to provide yourself the confidence that we have addressed this concern are:
> * Analogies with the elements of popular programming languages, such as C
> * More details on the definition of States
> * Expanded the definition of Commands with more examples
> * More details on the utility of Domain Theory in our setting
> * Further motivation for our theoretical excursion
>
> *Why use a partial function in equation 3?*
>
> The domain of a denotation is always a State and the codomain depends on the type of expression. For commands, the codomain is also a state. As some commands may not terminate, the denotation for them is undefined, i.e., we have a function which is not defined for some input arguments, hence a partial function.
>
> *There is some prior work [1] which indicates that more steps to find equilibrium points help with better OOD generalization. … DEAR penalizes longer trajectories... Is it possible to get improved performance if we ignore the potential conflict with domain theory?  ...*
>
> We will make sure to include the provided reference [1] in our paper. Furthermore, we would like to clarify that DEAR does not necessarily penalise longer trajectories. Regularisation is used only in the case we use alignment and even then the loss is normalised by the length of trajectory (see line 487). We are aware of the observation that more steps to find equilibrium points help with better OOD generalisation (reference [31] in our draft) and we used it with the alignment scheme.
>
> *What loss objective is used to train DEAR (e.g. cross entropy)?...*
>
> The loss function is algorithm specific as specified in the CLRS-30 paper. Thus motivated by this comment, we have emphasised (\emph)  and reworded L228-229 to make it clearer for any future readers: “Each task is independently learned, minimising the output loss plus any regularisation loss. The exact output loss is algorithm specific, therefore it can be either binary cross entropy or categorical cross entropy, cf. CLRS-30 for further details.”
>
> Once again we thank the reviewer for their insightful comments and excellent questions. We hope our reply addresses all concerns and questions, such that it may convince you to increase your score. We are of course happy to further engage with the reviewer for any remaining doubt.

---

> > ### Comment · Reviewer_1Bds · 2024-08-12
> >
> > Thank you for your detailed responses my questions. I hope the authors will include the new experiments as well as additional literature on domain theory as promised. I would like to retain my score.

---

> ### Author Response · Authors · 2024-08-13
>
> Thank you for taking the time to read our rebuttal. We are happy we have addressed all your questions through our detailed response, and will ensure that the updated experiments and additional literature on domain theory will be included in our paper. For these reasons, could we therefore kindly ask if there is any particular reason why you have not increased your score? We found your review insightful and it helped strengthen our paper. Please let us know if you have any further questions or suggestions.

---

### Author Rebuttal · Authors · 2024-08-06

We thank the reviewers for their thorough reviews and insightful comments. Each review has helped improve our paper by identifying areas that may have been misinterpreted and required further clarification. Here, we address the most important points which were raised by several of the reviewers.

*DEAR is a foundational model*

DEAR is a new equilibrium-based way to decide the termination both during training and inference in the setting of NAR. Using deep equilibrium models (DEQs) is motivated well theoretically, our empirical evidence suggests that equilibrium serves as an implicit bias towards better generalising solutions. While we refer to DEAR as a “model” in our paper, it is rather a *class of models* / *foundational model* as it can natively support different types of processors (Table 5; rebuttal PDF). DEAR targets specifically NAR, and any NAR application out there, e.g. Numeroso et al. (2023; ICLR), may benefit from it, but the purpose of the paper is to motivate the DEQ$\leftrightarrow$NAR relationship and integrate DEQs with NAR models.

*DEAR improves on the baselines and can achieve competitive results with highly advanced GNNs*

During the rebuttal period, we conducted new experiments showing DEAR is better than all other baselines that are of the same size/complexity class; this does not include Triplet-MPNN. This is especially exemplified on methods that are not given termination information at test time and also over methods that are given the ground-truth number of steps. After resolving anomalies with CLRS-30 (we give details below), DEAR with PGN outperforms comparable baselines and closely matches TripletMPNN with causal regularisation (2% difference).

If DEAR is used with Triplet-MPNN (Table 5), we achieve the best overall performance from all models.

Summary of new experimentations and updated results:
* We have added an overall model performance metric across all algorithms. DEAR outperforms baseline models of its size.
* We have shown that standard NAR models are fragile, w.r.t. changing the number of steps, showcasing that a good termination condition is essential to good performance.
* We compared a model that uses a dedicated NN layer to decide termination and we achieved a 5% overall improvement.
* We highlighted (by diamonds; see Table 1; rebuttal PDF) that our model *never* uses any ground-truth termination information – neither at train time nor at test time.
* We have included extreme out-of-distribution up to 32x larger sizes (Table 3; rebuttal PDF) tests with DEAR.
* We have included efficiency measures at those scales, showing that our model can improve inference speeds substantially, while still being performant (Table 4; rebuttal PDF).
* We have included Triplet-MPNN experiments as a baseline and combined with DEAR. We note that Triplet-MPNN is more computationally expensive, so we could manage to only train on a subset of the algorithms (Table 5; rebuttal PDF). We gave priority to those algorithms, that non-DEQ Triplet-MPNN improves over non-DEQ NAR, such as FW/DFS/etc.

*(Binary) Search anomalies*

**For those familiar with CLRS-30 (Veličković et al. 2022)**: The computation of the ground location in the official CLRS-30 implementation, which we use to generate the data, is slightly noisy.

In Binary Search, we aim to find the place to insert the target value `x` in the sorted array. Thus we need to point to the graph node that holds the smallest value in the array `A`, which is greater than `x`. However, if `x>max(A)` the answer is a pointer to the last value of the array, which by the convention used by CLRS-30 means we’d be inserting `x` at the wrong place. In other words, the answer to `A=[0.1, 0.2, 0.3] x=0.25` and `A=[0.1, 0.2, 0.3] x=0.35` is the same – insert x to the left of 0.3. This contributed some noise, so we fixed the sampler to always give `x` within `[0, max(A))`.

The other changes were to explicitly use `graph+pointer` instead of `node+mask_one` as the location and datatype of the pointer to the position in the array. This is something also done by Engelmayer et al. (2024). We also add an additional supervision signal, as done in Engelmayer et al. (2024), but at the output level rather than the hint level, since DEAR decouples algorithm iterations and solver iterations (L197-203; our paper draft).

**We have, of course, reran all models with this new search algorithm.** (Table 2; rebuttal PDF)

*Denotational Semantics*

Some of you raised the concern that Section 4 is hard to read and understand. As a result, we spent considerable time improving that part, and its corresponding appendix. We have added a new appendix dedicated to IMP. However, the NeurIPS rebuttal rules prevent us from being able to share the revised revisions, thus we opted to provide detail in individual replies by highlighting the changes we have made.


-----------


Summary of all the new changes:
* Highlighting key contributions of our paper:
  * The first robust equilibrium algorithmic reasoner; the model from the DEM blogpost performs terribly on the simplest task of BFS.
  * A regularisation scheme to encourage alignment with algorithm execution traces when training deep equilibrium neural algorithmic reasoners.
  * A comprehensive evaluation that shows DEAR is competitive to a model using a more expressive GNN.
  * Greatly improved efficiency – the speedup gains are sometimes as high as 50x (Table 4, rebuttal PDF) and we never exceed 0.5s/sample even at the most extreme scales.
  * Improved denotational semantics chapter & appendices, better connection with other sections and improved motivation of our theoretical excursion.
* The DEQ-NAR connection is formally motivated.
* Many new experiments showing integrating equilibrium with NAR results in strong models irrespective of the GNN architecture chosen as the processor.

---

### Decision · Program_Chairs · 2024-09-25

**Decision:**

Accept (poster)

**Comment:**

After carefully reading the reviews and rebuttal, as well as looking over the paper, I feel this work is ready for acceptance. The idea to use equilibrium-based approach to decide termination is interesting and of value to the community. I urge the authors to incorporate all the additional clarifications and details that they provided in the camera ready version of the paper.
I would also suggest punctuating the limitation of the method as well in an explicit paragraph in the camera ready, outlining the underlying assumption made by the approach and when it could fail (to the extend this is clear). In general, determining whether an algorithm terminates and when is an NP-complete problem, therefore it feels natural for there to be failure modes of the approach. But even without such a limitation section, the paper provides a new perspective on the problem that I think is very interesting and useful for the community!